# A Spark of Vision-Language Intelligence: 2-Dimensional Autoregressive Transformer for Efficient Finegrained Image Generation

**Liang Chen[1], Sinan Tan[2], Zefan Cai[3], Weichu Xie[4], Haozhe Zhao[1], Yichi Zhang[1]**
**Junyang Lin[2], Jinze Bai[2], Tianyu Liu[2], Baobao Chang[1]✉**
[1]Peking University   [2]Alibaba Group   [3]University of Wisconsin–Madison
[4]Beijing Institute of Technology    leo.liang.chen@outlook.com

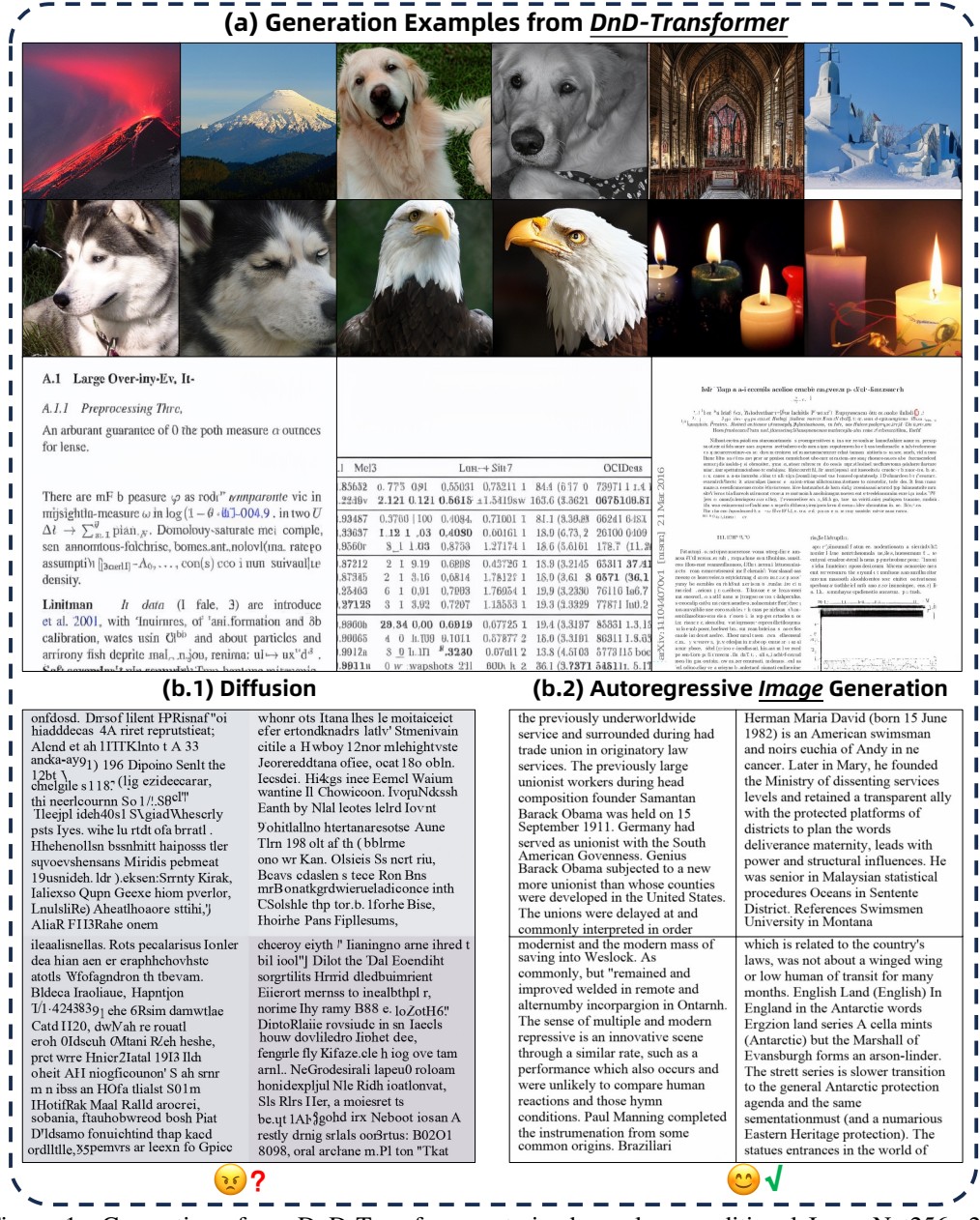

Figure 1: Generations from DnD-Transformers trained on class-conditional ImageNet256×256 (a.top) and unconditional arXiv images (a.bottom). Unconditional rich-text image generations by trained diffusion (b.1) and autoregressive model (b.2), where autoregressive model has dominating performance, showing a spark of vision-language intelligence after purely training on images.

ABSTRACT

This work tackles the information loss bottleneck of vector-quantization (VQ) autoregressive image generation by introducing a novel model architecture called the 2-Dimensional Autoregression (DnD) Transformer. The DnD-Transformer predicts more codes for an image by introducing a new direction, *model depth*, along with the sequence length. Compared to 1D autoregression and previous work using similar 2D image decomposition such as RQ-Transformer, the DnD-Transformer is an end-to-end model that can generate higher quality images with the same backbone model size and sequence length, opening a new optimization perspective for autoregressive image generation. Furthermore, our experiments reveal that the DnD-Transformer's potential extends beyond generating natural images. It can even generate images with rich text and graphical elements in a self-supervised manner, demonstrating an understanding of these combined modalities. This has not been previously demonstrated for popular vision generative models such as diffusion models, showing a spark of vision-language intelligence when trained solely on images. Code, datasets and models are open at https://github.com/chenllliang/DnD-Transformer.

## 1 INTRODUCTION

The field of autoregressive (AR) image generation is experiencing a resurgence of interest, largely driven by groundbreaking advancements in large language models (LLMs), exemplified by the release of ChatGPT (OpenAI, 2022). Because typical AR image generation methods also predict output in a next-token prediction manner, this resemblance has sparked significant efforts in two main areas: 1) transferring advanced, large-scale training techniques and expertise from LLMs to AR image generation models (Bai et al., 2023; Tian et al., 2024; Sun et al., 2024), and 2) developing truly multimodal foundation models capable of both understanding and generating multimodal information within a unified training framework (Lu et al., 2022; 2023; Team, 2024). These developments have the potential to lead to more versatile and powerful multimodal AI systems.

A review of the development history of AR image generation approaches reveals significant efforts focused on finding better sequential decompositions of images and balancing reconstruction fidelity with prediction difficulty. Early models, like PixelCNN (van den Oord et al., 2016), generated images pixel by pixel. This approach was later enhanced by using vector-quantized variational autoencoders (VQVAEs) to compress images and model the prior distribution of discrete tokens in a compact latent space (Van Den Oord et al., 2017). Vector quantization (VQ) paved the way for notable models such as VQGAN (Esser et al., 2021), DALL·E (Ramesh et al., 2021), and MUSE (Chang et al., 2023), and it remains a core technique in recent AR image generation models like VAR (Tian et al., 2024) and LlamaGen (Sun et al., 2024), and multimodal foundation models like LVM (Bai et al., 2023), Unified-IO (Lu et al., 2022; 2023), and Chameleon (Team, 2024).

However, despite advancements in AR image generation, VQ-based autoregressive methods face two persistent criticisms, especially juxtaposed with latent diffusion models (Rombach et al., 2022):

**1) Information loss inherent in the quantization process.** Quantization, specifically in VQVAE, introduces significant information loss. With a typical configuration (N=8192, f=16), the Information Compression Ratio (ICR $= \frac{\log_2 N}{24f^2}$, explained in Equation 1) is just 0.21%, drastically lower than the 8.3% of Stable Diffusion's VAE[1], hindering fine-grained detail reconstruction. According to Chameleon (Team, 2024), the authors note that their VQ tokenizer struggles to reconstruct finegrained details like text in images, which we believe is due to the low ICR of their tokenizer.

**2) Substantially increased computational requirements for producing higher-quality images.** According to Equation 1,Increasing ICR by expanding the latent space (N) is logarithmically limited

---

[1]The Stable Diffusion VAE (https://huggingface.co/stabilityai/sd-vae-ft-mse) uses a downsampling factor (f) of 8 and 4 channels, with fp32 tensor precision ($\log N = 4 \times 32$).

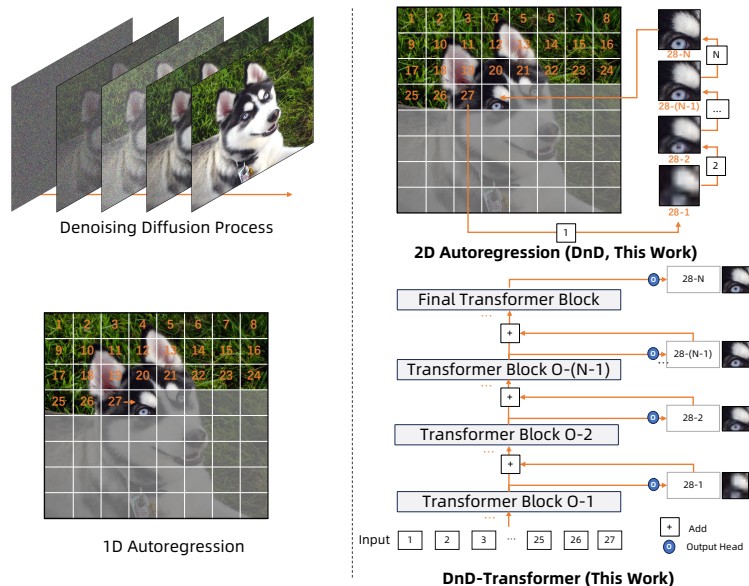

Figure 2: Illustration of the proposed DnD-Transformer. N denotes the number of depth autoregression. O-i denotes the transformer layer index for the i-th prediction head. Each transformer layer predicts the corresponding depth code, achieving multi-code prediction within one forward pass.

and computationally expensive leading to potential codebook collapse and more embedding parameters, while reducing the downscaling factor (f) significantly increases computational overhead due to a longer token sequence of $O(1/f^2)$ and a higher transformer computation complexity of $O(1/f^4)$.

We draw inspiration from the Residual Quantization method (Lee et al., 2022b), which provides a new dimension for sequentially decomposing the image for better generation quality. However, the proposed RQ-Transformer employs two separate transformer models. This structure presents difficulties in integrating current LLMs for end-to-end training. In this work, we aim to solve the problem covering the two mentioned concerns: ***Can we overcome the information loss of VQ-based AR image generation without increasing overall computation budget in an end-to-end manner?***

We propose a novel paradigm for AR image generation called 2-Dimensional Autoregression (DnD) and DnD-Transformer, an end-to-end model architecture. DnD Autoregression introduces a new depth dimension along with the original spatial dimension. In the depth dimension, the image patch could be decomposed in any causal coarse-to-fine order, including the residual decomposition (Lee et al., 2022b), Gaussian denoising decomposition (Ho et al., 2020) and etc. With a depth of $d$ and other configurations unchanged, the ICR of DnD Autoregression becomes $d \times \frac{\log N}{24f^2}$, more effectively reducing the information loss comparing to increasing the codebook size $N$.

The remaining problem is how to predict the $d$ times more tokens effectively. We propose the DnD-Transformer. As shown in Figure 2, it inserts multiple prediction heads into the backbone transformer decoder model to predict the depth codes and conduct additional autoregressive predictions in each forward process. Different from RQ-Transformer (Lee et al., 2022b), the DnD-Transformer does not require additional modules or increased sequence length, making it applicable to any language model architecture and efficiently generate more fine-grained images.

Our experiments show several interesting results: 1) superior reconstruction of fine-grained image details using residual image decomposition in VQVAEs, disproving VQ's limitations with text-rich images; 2) more efficient and lower-entropy decomposition with DnD autoregression compared to 1D methods, evidenced by lower training cross-entropy loss despite predicting more codes; 3) significant outperformance of the AR baseline on ImageNet 256x256 generation, achieving up to 1.54 FID and 82.6 IS improvements (XXL model, cfg=2) without increased model size or sequence length, even surpassing larger LlamaGen model trained with longer sequence length; and 4) DnD-Transformer shows that we can conduct accurate language modeling with pure image generation model outperforming diffusion models like DDPM and Stable Diffusion on dedicated rich-text image datasets, highlighting the distinct advantage of autoregressive models for multimodal modeling.

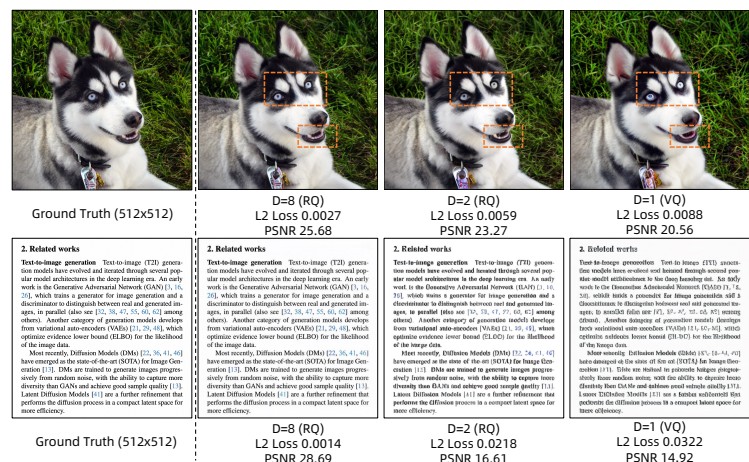

Figure 3: Performance of our visual tokenizers of different depths. The reconstruction of complex features (i.e., eyes, mouse and text) gains significant improvement as the depth increases.

## 2 2D VISUAL TOKENIZER AND 2D AUTOREGRESSION

### 2.1 UNDERSTAND VQVAE AS COMPRESSION

We introduce the basics of AR generation in Section A in the appendix. We can better understand the reconstruction ability of VQVAE from the lens of compression. Let us assume a VQVAE with downscaling factor $f$, codebook size $N$, input image's size of $H \times W$, then the shape of the quantized code is $h \times w = (H/f) \times (W/f)$. We assume that the code follows a uniform distribution, so each code has $\log_2 N$ bits information. Its information compression ratio (ICR) is as follows.

$$ICR(N, f) = \frac{(H/f) \times (W/f) \times \log_2 N}{H \times W \times 3 \times \log_2 256} = \frac{\log_2 N}{24 f^2} \quad (1)$$

A typical configuration (N=8192, f=16) results in 0.21% ICR. As a comparison, JPEG typically achieves a far larger compression ratio from 5% to 10%, resulting in minimal perceptible loss in image quality (Haines, 1992). To increase ICR, the 1D AR method could increase $N$ (might face the codebook collapse problem (Mentzer et al., 2023) and the improvement is logarithmically bounded) or decrease $f$ (more effective, but increases the token count quadratically).

### 2.2 IMAGES' 2D DECOMPOSITION AND QUANTIZATION

As pointed out by Equation 1, the information compression ratio of VQVAE is bounded by the size of the codebook and the downscaling ratio. Residual Quantization (Lee et al., 2022b) proposes a new direction to quantize the image feature with multiple residual codes to reduce the quantization error and improve the quality of the reconstruction. For a feature map having $h \times w$ vectors, RQVAE uses $h \times w \times d$ codes to quantize the feature map, where $d$ is the depth dimension of the code. For each feature vector $\mathbf{v}$, RQ finds $d$ codes $(q_1, q_2, ..., q_d)$ by sequentially conducting $d$ times residual decomposition and quantization operation $\mathcal{Q}(x)$ as finding the closest entry to $x$ from the codebook:

$$q_d = \mathcal{Q}(r_{d-1}), \quad r_d = r_{d-1} - q_d, \quad r_0 = \mathbf{v} \quad (2)$$

Consequently, the sum of the residual codes $\sum_{i=1}^{d} q_i$ is expected to approximate more closely the feature vector $\mathbf{v}$, thus reducing the quantization error. We generalize this process as two-dimensional autoregression (DnD), which extends beyond Markov residual decomposition and can be applied to any decomposition operation, such as the diffusion process (Ho et al., 2020), etc.

DnD Autoregression quantizes a 2D feature map $\mathbf{m} \in \mathcal{R}^{h \cdot w \cdot c}$ by decomposing it in two directions. First, $\mathbf{m}$ is divided into $h \cdot w$ feature vectors. Second, each vector $\mathbf{v}$ is decomposed into $n$ codes $(q_1, ..., q_n)$ using a function $\mathcal{D}^n(\mathbf{v}, \mathcal{Q})$ based on a codebook $\mathcal{Q}$. The resulting quantized map $\mathbf{q}$ has shape $h \cdot w \cdot n$ and is predicted in depth-first-spatial-second order. This decomposition could

| Depth(RQVAE) | ImageNet 256×256 | | |
| --- | --- | --- | --- |
| | rFID↓ | L2 Loss↓ | Code Usage↑ |
| 1 | 2.98 | 0.11 | 100% |
| 2 | 0.93 | 0.08 | 100% |
| 4 | 0.60 | 0.05 | 100% |
| 8 | 0.42 | 0.04 | 100% |
| SDXL | 0.68 | 0.05 | - |
| SD3 | 0.67 | 0.04 | - |

(a) **Reconstruction Performance on ImageNet 256×256 Validation Set.**

| Depth(RQVAE) | Text256 | Text512 | arXiv512 |
| --- | --- | --- | --- |
| | | rOCR↑ | |
| 1 | 0.15 | 0.73 | 0.14 |
| $1^{\dagger}$ | 0.00 | 0.00 | 0.00 |
| 2 | 0.50 | 0.81 | 0.49 |
| 8 | 0.80 | 0.83 | 0.67 |
| SDXL | 0.72 | 0.83 | 0.66 |
| SD3 | 0.82 | 0.83 | 0.74 |

(b) **Reconstruction OCR Performance.** † indicates zero-shot tokenizer trained on ImageNet.

Table 1: **Ablation studies on the reconstruction performance of visual tokenizers**. Our trained tokenizers all have a $f = 16$ downscaling factor and $N = 16384$ codebook size.

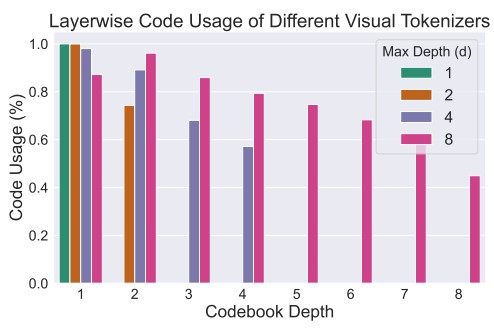

(a) Layerwise code usage of visual tokenizers.

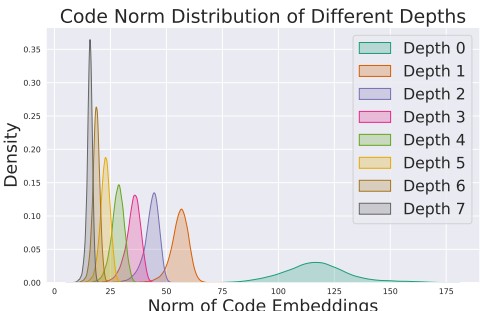

(b) Code Norm Distribution for Tokeniers

Figure 4: Analysis of visual tokenizers.

also be non-Markov, unlike RQVAE. The selection of potentially better decomposition functions is left for future exploration. We use RQ-VAE (Lee et al., 2022b) as the implementation of 2D decomposition method in our experiments. We still use the residual quantization from Equation 2 as $\mathcal{D}^n$. DnD decomposition increases the ICR $d$ times (Equation 3), more effectively than increasing codebook size. The remaining challenge of predicting $d$ times more codes is addressed by our DnD-Transformer.

$$ICR(N, f, d) = d \times \frac{(H/f) \times (W/f) \times \log_2 N}{H \times W \times 3 \times \log_2 256} = d \times \frac{\log_2 N}{24 f^2} \tag{3}$$

## 2.3 RECONSTRUCTION PERFORMANCE

We evaluate the reconstruction performance of our trained visual tokenizers with varying maximum codebook depths using the standard ImageNet dataset as the benchmark. All images are resized to 256×256 resolution. We train the different visual tokenizers using the same training objectives as in Lee et al. (2022b), and assess the reconstruction Fréchet Inception Distance (rFID) on the ImageNet validation set using ADM's evaluation suite (Dhariwal & Nichol, 2021). The results are presented in Table 1a. For comparison, we include the rFID from the VAE of SDXL (Podell et al., 2023) and Stable-Diffusion 3 (Esser et al., 2024) . Our findings demonstrate that our trained visual tokenizer achieves an rFID lower than 1 with two or more codebook depths, even surpassing the performance of SD3's continuous VAE with less theoretical information loss. As shown in the example from Figure 3, by increasing code depth, we could reconstruct more fine-grained details in the image.

**Code Usage.** We further analyze the code usage in each codebook layer, with results shown in Figure 4a. The analysis indicates that usage generally decreases as depth increases. This is due to the diminishing diversity of code usage as the residual decomposition progresses deeper, resulting in smaller feature norms and more centralized code usage according to Figure 4b. Interestingly, we do not observe signs of codebook collapse with the DnD visual tokenizers, even when using a large

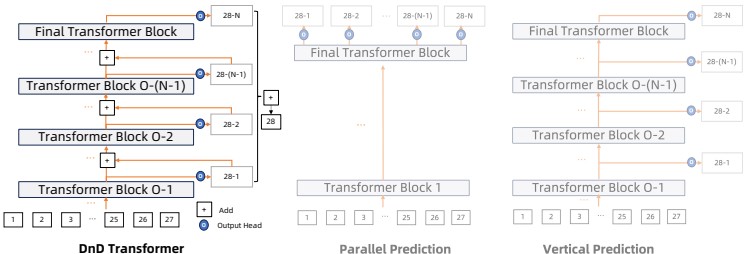

Figure 5: Different explored multi-token prediction architectures for DnD-Transformer, which are all designed to generate multiple codes with one forward pass.

codebook size (16384), as mentioned in previous work (Mentzer et al., 2023). While they reported much lower code usage ($< 50\%$), our tokenizer achieves $100\%$ usage across all maximum depths.

## 2.4 VQVAES CAN PERFECTLY RECONSTRUCT RICH-TEXT IMAGES

A prevalent criticism of VQVAE has been its alleged intrinsic information loss problem, particularly its inability to reconstruct images with fine details, such as those containing rich text (Team, 2024). However, we argue that this claim is unfounded. Our findings suggest that VQVAE can indeed achieve perfect reconstruction of detailed images, when provided sufficient data and an increased number of codes used to represent each image. This demonstrates that the perceived limitations of VQVAE can be overcome through appropriate data-centric adjustments and model scaling-up.

**rOCR - A New Metric.** We proposes rOCR, a novel metric for evaluating rich-text image reconstruction. Unlike rFID/L2 Loss, rOCR measures textual recognizability using the Qwen2-VL-72B (Wang et al., 2024a) visual language model for OCR. The metric computes the Rouge-L score between recognized and groundtruth text (or original image OCR if groundtruth is unavailable).

**Experiments and Results.** Two rich-text image datasets, Text-Image and arXiv-Image (details in Section 4.1), were used to train visual tokenizers. Performance (rOCR scores) was evaluated on both datasets' 1K test sets, compared against ImageNet-trained tokenizers, SDXL's (Podell et al., 2023) and Stable-Diffusion-3's VAE (Esser et al., 2024). Text-Image was also tested at a reduced 256×256 resolution to assess resolution impacts. Table 1a shows the rOCR results, with reconstruction examples in Figures 3 and 9. Results indicate more training data and deeper tokenizers improve text reconstruction. Unlike Team (2024), our discrete visual tokenizers excel in rich-text image reconstruction even compared to continuous VAEs.

## 3 THE DND-TRANSFORMER

Prior section showed DnD visual tokenizers effectively reconstruct fine details like text. However, efficiently predicting the increased number of depth codes ($d$ times more) remains challenging. Existing methods, like RQ-Transformer, use a separate transformer for depth, hindering integration with LLMs. We propose an efficient end-to-end architecture for multi-code prediction.

### 3.1 DND-TRANSFORMER DESIGN

Figure 5 shows DnD-Transformer and its variants: Parallel and Vertical Prediction. Parallel Prediction adds multiple prediction heads for simultaneous multi-depth code prediction, similar to accelerated LLM inference (Cai et al., 2024). However, this ignores the coarse-to-fine nature (Figure 4b) of code distributions, where deeper codes have smaller norms and are more centered. Vertical Prediction addresses this by sequentially predicting codes. Adding autoregression further refines this by conditioning deeper code predictions on previous ones, achieving the best multi-layer code prediction without increasing model parameters or sequence length. Ablation on the structure design is shown in Table 3 from Appendix.

## 3.2 IMPLEMENTATION DETAILS

As shown in Figure 5, the DnD-Transformer enhances the vanilla transformer decoder by adding output heads and embedding addition operation. Assuming the linearized codemap's length is $L = h \times w$ and code depth is $d$, the DnD-Transformer performs $L$ forward passes, generating $d$ codes sequentially during each pass. After generating for all depths, the embeddings are summed to form the next input token. This allows the model to produce $L \times d$ tokens with just $L$ forward passes, improving generation quality without increasing inference costs. The only additional hyper-parameter is the layer indexes for codes at different depths. We use the same architecture as the LLaMA (Touvron et al., 2023) transformer decoder; training details can be found in Appendix E.

## 4 EXPERIMENTS AND FINDINGS

### 4.1 TASKS AND DATASETS

**Class-Conditional Image Generation.** We conduct standard conditional image generation task with ImageNet-1k benchmark. Images are resized to 256×256 resolution during training and evaluation. We sample 50k images with classes uniformly distributed, and compute the FID, IS, Precision and Recall aganist the training set data using the ADM evaluation tool Dhariwal & Nichol (2021).

**Unconditional Rich-Text Image Generation.** We collect two datasets for this task. Dataset examples are shown in Figure 10 in the Appendix. Models are trained in a unconditional setting in this task. We aim to explore whether the tested vision generation models could understand and generate the complex logical interrelation among the generated elements such as language.

1. *Pure Text Images (Text-Image).* The dataset is automatically rendered from a portion of English wikipedia (Foundation), consisting of 2.4M images. Each image has a original resolution of 512×512 and a font size of 32pt. We set a maximum of 100 words in each image with a paddling margin of 20pt. We use the PILLOW library to render the image.

2. *arXiv Images (arXiv-Image)* we first download the papers in PDF format from arXiv.org, and render the pages to image of A4 resolution ($1260 \times 1782$) with PDF2IMAGE tool. We then randomly crop ten 512×512 image from each pages and finally collect 2M images.

We developed an evaluation pipeline that combines Optical Character Recognition (OCR) and Perplexity Measurement to assess the quality of generated images, focusing on their textual content. First, we use the state-of-the-art Vision-Language Model Qwen2-VL-72B to extract text from the images. Then, we calculate the text's perplexity using the Qwen2.5-72B model, treating it as the evaluator. The resulting score, $PPL_{ocr}$, is compared to the score of ground truth data from the training images to establish a performance upper bound.

### 4.2 MODELS

**Visual Tokenizers.** We train our visual tokenizer based on RQVAEs (Lee et al., 2022b). We train tokenizers with code depths of $\{1, 2, 4, 8\}$ and scaling factor $f = 16$ across different experiments. We choose the checkpoint with best rFID across 150 epochs. Performance comparison of different visual tokenizers is shown in Table 1. We follow Lee et al. (2022b) to train the visual tokenizers. Details of the training of visual tokenizers are listed in Appendix B. Reconstruction performance of the trained visual tokenizers is shown in Table 1.

**DnD-Transformer.** We train two size of DnD-Transformers across our experiment, namely DnD-Transformer-XXL (1.4B) and DnD-Transformer-XXXL (2.5B). Basically, DnD-Transformer inherits the LLaMA (Touvron et al., 2023) architecture. The XXL version strictly align with the LlamaGen-XXL baseline to be fairly compared. Details of the model are shown in Appendix E.

**Implemented Baselines for Class-Conditional Image Generation.** LlamaGen (Sun et al., 2024) is the major baseline and state-of-the-art model for AR image generation on ImageNet. Our implemented code primarily refers to the same training codebase for fair comparison. LlamaGen could be also viewed as a special version of DnD-Transformer where the decomposition depth equals to 1.

| Type | Model | #Para. | FID↓ | IS↑ | Precision↑ | Recall↑ |
|------|-------|--------|------|-----|-----------|---------|
| Diffusion-Reported | ADM (Dhariwal & Nichol, 2021) | 554M | 10.94 | 101.0 | 0.69 | 0.63 |
| | CDM (Ho et al., 2022) | – | 4.88 | 158.7 | – | – |
| | LDM-4 (Rombach et al., 2022) | 400M | 3.60 | 247.7 | – | – |
| | DiT-XL/2 (Peebles & Xie, 2023) | 675M | 2.27 | 278.2 | 0.83 | 0.57 |
| AR-Reported | VQGAN (Esser et al., 2021) | 1.4B | 5.20 | 280.3 | – | – |
| | RQTransformer (Lee et al., 2022a) | 3.8B | 7.55 | 134.0 | – | – |
| | LlamaGen-XXL (cfg=2) (Sun et al., 2024) | 1.4B | 3.64 | 296.5 | 0.86 | 0.51 |
| | LlamaGen-XXL† (384×384, cfg=2) (Sun et al., 2024) | 1.4B | 2.52 | 295.4 | 0.84 | 0.56 |
| | LlamaGen-3B (cfg=2) (Sun et al., 2024) | 3.1B | 4.21 | 325.2 | 0.87 | 0.49 |
| | LlamaGen-3B† (384×384, cfg=2) (Sun et al., 2024) | 3.1B | 2.81 | 311.6 | 0.84 | 0.54 |
| | VAR (Tian et al., 2024) (with reject sampling) | 2.0B | 1.73 | 350.2 | 0.82 | 0.60 |
| | HQ-Transformer (You et al., 2022) (with reject sampling) | 1.4B | 4.35 | - | 0.73 | 0.55 |
| | MAR (Li et al., 2024) (trained longer, 400 epochs) | 400M | 1.98 | 290.3 | - | - |
| AR-Implemented | LlamaGen-XXL (cfg=4) | 1.4B | 7.67 | 345.1 | **0.89** | 0.35 |
| | LlamaGen-XXL (cfg=2) | 1.4B | 4.12 | 266.9 | 0.83 | 0.49 |
| | DnD-Transformer-XXL (cfg=4) | 1.4B | 6.55 | **427.7** | **0.89** | 0.42 |
| | DnD-Transformer-XXL (cfg=2) | 1.4B | 2.58 | 295.6 | 0.83 | 0.56 |
| | DnD-Transformer-XXL (cfg=1.7) | 1.4B | 2.78 | 239.2 | 0.82 | 0.56 |
| | DnD-Transformer-XXL (cfg=1.5) | 1.4B | 2.96 | 232.5 | 0.80 | 0.57 |
| | DnD-Transformer-XXXL (cfg=4) | 2.5B | 6.48 | 413.0 | **0.89** | 0.42 |
| | DnD-Transformer-XXXL (cfg=2) | 2.5B | 2.77 | 319.1 | 0.85 | 0.54 |
| | DnD-Transformer-XXXL (cfg=1.7) | 2.5B | **2.21** | 279.3 | 0.83 | **0.58** |
| | DnD-Transformer-XXXL (cfg=1.5) | 2.5B | 2.52 | 244.2 | 0.80 | 0.59 |

Table 2: **Model comparisons on class-conditional ImageNet 256×256 benchmark**. The "Reported" results refer to Sun et al. (2024). The "Implemented" results are conducted in this work. † indicates that the model is unorthodoxly trained at 384×384 resolution, which requires 2.25 times longer sequence length compared to our implemented models. "cfg" means the scale of classifier-free guidance. The number of depth autoregression is 2 for DnD-Transformers.

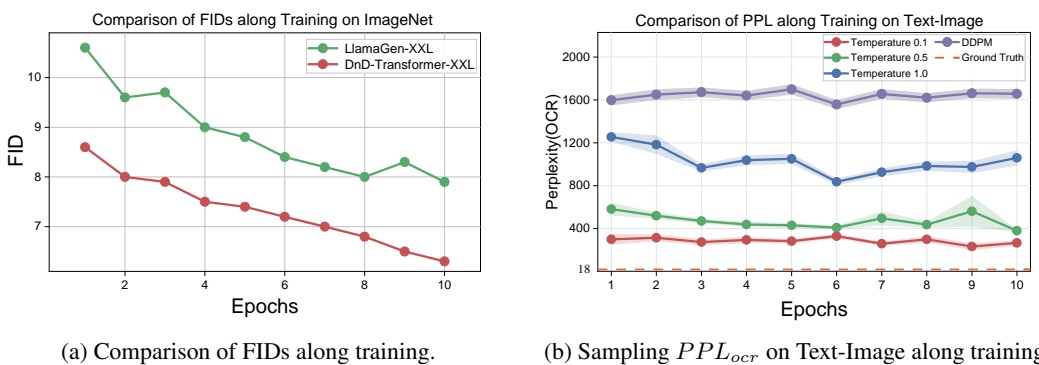

(a) Comparison of FIDs along training.

(b) Sampling $PPL_{ocr}$ on Text-Image along training.

Figure 6: Curves during training.

**Implemented Baselines for Rich-Text Image Generation.** We select multiple diffusion models as the baselines, including DDPM (Ho et al., 2020), Stable Diffusion XL (SDXL) (Podell et al., 2023) and Stable Diffusion v3.0 (SD3) (Esser et al., 2024). For DDPM, we train the model on the dataset from scratch. For SDXL and SD3, we finetune the checkpoints from the official websites.

## 4.3 RESULTS OF CLASS-CONDITIONAL IMAGE GENERATION

As demonstrated in Table 2, our DnD-Transformer significantly outperforms the 1D autoregressive baseline LlamenGen across various scales and generation evaluation metrics, including FID and IS. This superior performance is achieved while maintaining the same number of parameters in the backbone model, based on our reported and implemented results. It is noteworthy that our 2.5B model, trained with a sequence length of 256, even outperforms the 3.1B LlamaGen model, which was trained with a much longer image sequence length of 576. This result demonstrates that the DnD-Transformer can effectively predict a greater number of tokens within a shorter sequence length, highlighting its significant potential to revolutionize the one-dimensional autoregressive paradigm. We randomly sample some generation results as shown in Figure 1 and compare the generation performance with 1D-AR in Figure 11,12 and 13 from the Appendix. The comparative analysis clearly illustrates the effectiveness of our approach to generate high-quality images.

## 4.4 RESULTS OF RICH-TEXT IMAGE GENERATION

**Generation Results on Text-Image.** A DnD-Transformer (depth 1) and a DDPM model were trained on the same text-image dataset. Comparing 250 randomly sampled images from each, the AR model significantly outperformed the diffusion model in generating coherent text (lower OCR

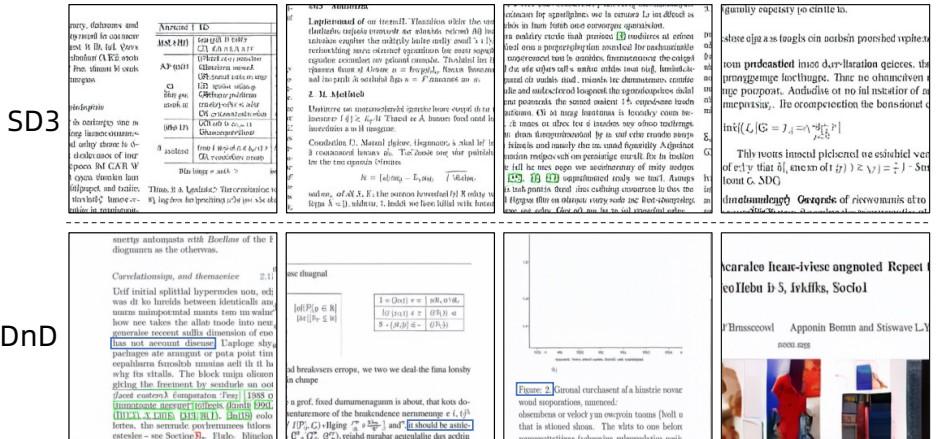

Figure 7: Comparison of Unconditional Rich-Text Image Generation on the more complex arXiv-Image dataset. SD3 is hard to generate valid words, while DnD-Transformer demonstrates an ability to generate semantically appropriate phrases, as marked in blue. More baselines are in Figure 14.

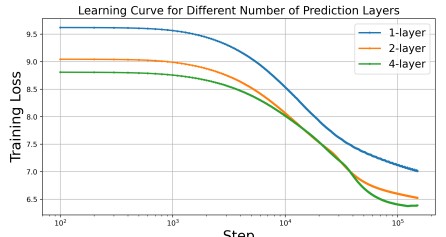
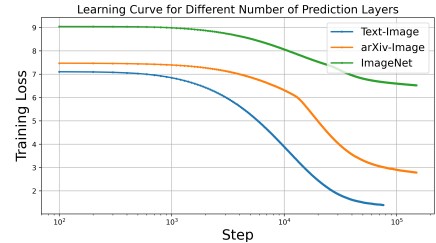

(a) Training Loss for DnD-Transformer trained with different number of prediction heads.

(b) Training Loss when trained on different domain datasets.

Figure 8: Analysis of code depths and domains during training DnD-Transformers.

perplexity 6b; Generation examples 1, 15, 16, 17 and 18 ). This suggests the AR model's discrete token reconstruction enables effective autoregressive modeling. We also find that with a lower sampling temperature, the model would generate text images with lower PPL just like LLMs. Conversely, the diffusion model's simultaneous generation hinders text coherence.

**Generation Results on arXiv-Image.** An 8-layer visual tokenizer and corresponding DnD-Transformer trained on arXiv-Image outperformed diffusion model baselines, generating more valid words and phrases (Figure 7). However, arXiv-Image generation lagged behind Text-Image generation, suggesting joint language and figure modeling is more challenging. More results and baselines are in Figure 14 and 19. While SD3's VAE reconstructs arXiv images well (Table 1b), its generative performance is inferior to DDPM and AR, suggesting its latent space is less suitable for language modeling comparing to pixel or discrete space.

**A Spark of Vision-Language Intelligence.** Autoregressive (AR) image generation exhibits a marked advantage over diffusion models in producing text-rich images, as demonstrated by our results. The pixel-level language generation inherent to AR models facilitates this capability. Despite limitations imposed by our current training data and model size (preventing direct comparison with large language models), these findings suggest a promising pathway towards vision-language intelligence where **language understanding emerges directly from visual perception.** Furthermore, our pure image learners display behaviors mirroring language model issues such as repetition and hallucination (Figure 20), implying the potential for integrating pure language modeling into a unified autoregressive framework for joint vision-language image modeling.

### 4.5 TRAINING BECOMES EASIER WHEN PREDICTING MULTIPLE CODES, SAMPLING NOT

Deeper DnD-Transformer codes achieve lower cross-entropy loss during training (Figure 8a), indicating lower entropy image decompositions. However, despite this, increased depth doesn't improve

ImageNet generation fidelity, possibly due to the larger sampling space. Exploring this multi-depth sampling space for better generation is a promising research direction.

### 4.6 AR TRAINING LOSS FOR DIFFERENT DOMAINS ALIGN WITH INNER RANDOMNESS

Training loss for the same DnD-Transformer varies significantly across datasets (Figure 8b), being notably higher for ImageNet than rich-text images. While rich-text image loss nears that of LLMs, ImageNet loss sits between text and natural image datasets. The AR model's LLM-like training suggests it learns language from visual input alone, implying language's visual representation has lower entropy than natural images, easing the learning process.

## 5 RELATED WORK

**Image Generation with VQVAE.** The vector quantization (VQ) method has been pivotal in the development of generative models (Chen et al., 2024; Ramesh et al., 2021; Yu et al., 2022; Chang et al., 2023), which achieve image generation through the prediction of discrete image tokens. Efforts in this area focus on two main directions: the optimization of image tokenization techniques (Esser et al., 2021; Mentzer et al., 2023; Yu et al., 2023; 2024; Weber et al., 2024; You et al., 2022), and the strategic planning of effective decompositions of image tokens, such as MaskGit (Chang et al., 2022) and VAR (Tian et al., 2024) or incorporating a diffusion loss such as MAR (Li et al., 2024). Meanwhile, alongside the advancement of large language models, there is growing interest in autoregressive image generation, which predicts image tokens sequentially (Tian et al., 2024; Sun et al., 2024). Recent research has also focused on developing multimodal foundation models (Lu et al., 2023; Kondratyuk et al., 2024; Wang et al., 2024b) that integrate both understanding and autoregressive image generation capabilities. They typically convert images or videos into sequences of discretized tokens and train over combined text-image/video token sequences within the AR modeling framework (Lu et al., 2022; Bai et al., 2023; Xie et al., 2024; Team, 2024). However, these models struggle with inherent information loss during the image quantization and the significantly increased computational demands when generating higher-quality images.

**Rich-Text Image Generation.** Despite recent significant progress in image generation, the task of rich-text generation within images remains a persistent challenge (Chen et al., 2023b; Ma et al., 2024; OpenAI, 2024). Most advancements have been witnessed in diffusion models (Betker et al., 2023; Saharia et al., 2022b;a), these models either leverage large language models to enhance the character spelling capabilities of generative models (Saharia et al., 2022b; Balaji et al., 2023; Saharia et al., 2022a) or attempt to explicitly control the position and content of the text using additional supervision from different modules (Tuo et al., 2024; Yang et al., 2023; Liu et al., 2024). However, most diffusion-based methods have primarily focused on text rendering Chen et al. (2023a;b); Balaji et al. (2023); Saharia et al. (2022a) in image generation, often limited to generating short words for logos and posters (Yang et al., 2023; Ma et al., 2023; 2024). The full potential of rich-text image generation remains largely unexplored. Our methods, which build on the foundation of DnD Autoregression, show substantial progress in generating rich-text images in an unconditional manner, highlighting the feasibility of conducting joint vision-language modeling tasks using purely images.

## 6 CONCLUSION

This paper investigated the limitations of autoregressive (AR) image generation methods, particularly the information loss and computational burden associated with vector quantization (VQ). We introduced 2-Dimensional Autoregression (DnD) and a novel end-to-end architecture, DnD-Transformer, which leverages a depth dimension autoregression alongside the spatial dimension to mitigate these limitations. Our experiments demonstrate that DnD-Transformer achieves significant improvements in image quality, outperforming strong baselines like LlamaGen without increasing model size or sequence length. Notably, DnD-Transformer showcases emergent vision-language intelligence, generating text-rich images unconditionally, a known weakness of diffusion models. These findings highlight the potential of DnD for efficient and high-quality AR image generation and underscore the promise of this approach for advancing multimodal foundation models.

## 7 ACKNOWLEDGMENTS

We thank all reviewers for the valuable advice. This work is supported by the National Science Foundation of China under Grant No.61876004.

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

## A  PRELIMINARY: AUTOREGRESSIVE IMAGE GENERATION

In this section, we introduce the fundamentals of autoregressive image generation. The pipeline is rooted in the Vector Quantized Variational Autoencoder (VQVAE) (Van Den Oord et al., 2017) and the autoregressive Transformer (Vaswani et al., 2017). This approach has been adopted from the early DALLE (Ramesh et al., 2021) to the latest LlamaGen (Sun et al., 2024).

### A.1  STEP1: TRAIN THE VISUAL TOKENIZER AND TOKENIZE THE IMAGES

Images initially exist in the pixel-level RGB color space, which consists of little semantic information and makes it challenging to directly model prior knowledge. For example, an image with a resolution of $256 \times 256$ comprises $256 \times 256 \times 3 = 196,608$ distinct values, representing the individual red, green, and blue intensities for each pixel. The large sequence length makes it difficult to train in autoregressive manner similar to language models' technique. Van Den Oord et al. (2017) proposed the Vector Quantized Variational Autoencoder (VQVAE), which significantly alleviates the problem. It downscales and tokenizes the image from the original sparse RGB space into a dense and discrete representational space (codebook) $\mathcal{Q}$ by finding the nearest entry. The VQVAE is typically implemented in an encoder-decoder architecture, with its primary training objective being to minimize the image reconstruction loss. You could refer to Van Den Oord et al. (2017) for details in training a standard VQVAE.

### A.2  STEP2: LEARN THE PRIOR DISTRIBUTION OF IMAGE TOKENS

Having tokenized the source images into discrete tokens and trained a visual decoder to map these tokens back to real images, the next crucial step is to learn the prior distribution of the discrete tokens. This distribution enables the sampling process, which is essential for generating new images. AR Image generation generally first linearizes the $h \times w$ image tokens $q \in \mathcal{Q}$ in a raster scan order and formalize 1D sequence $(q_1, q_2, q_3, ..., q_{h \times w})$ for the transformer (Vaswani et al., 2017) model to learn.

During training, the training objective is the same as GPT's next token prediction task (Radford et al., 2018), that the model is required to predict the next image token given the previous tokens and class or text conditional tokens $\prod_{t=1}^{h \times w} p(q_t \mid q_{<t}, c)$. After training, we can generate images by autoregressively sampling $h \times w$ tokens from the model. The sampled 1D sequence of image tokens is then reshaped to 2D code map with height $h$ and width $w$. This reshaped token map is subsequently fed into the trained VQVAE decoder, which reconstructs the final image from the code representation.

**Classifier-Free Guidance** As a technique to enhance the visual quality and text-image alignment, classifier-free guidance (Ho & Salimans, 2022) has been adopted across the diffusion models (Rombach et al., 2022; Podell et al., 2023), VQ models (Chang et al., 2023) and autoregressive models (Sun et al., 2024) for image generation. During the training, the model is exposed to data with and without conditioning: the conditioning is randomly discarded from a fraction of the training samples. We have implemented this approach in our model as well. Specifically, during training, we randomly replace the conditional embedding with a learnable unconditional embedding in 10% of the cases. At the inference stage, the logits $\ell_g$ are recalculated for each generated token. We form the $\ell_g$ by subtracting the unconditional logits $\ell_u$ by conditional logits $\ell_c$ with the guidance scale $t$ through the following equation:

$$\ell_g = \ell_u + (\ell_c - \ell_u) \times t \tag{4}$$

## B  TRAINING DETAILS OF VISUAL TOKENIZERS

We follow (Lee et al., 2022b) to train the 2D tokenizers with residual decomposition a combined objective of l2 loss, GAN loss and perceptual loss. Codes from different depth share the same codebook. We train all tokenizers a fixed learning rate of 4e-5, a total batch-size of 256 for 100 epochs and select the one with lowest validation loss as the final tokenizers. We conduct all training on 8×A100 GPUs.

## C  RECONSTRUCTION RESULTS OF TEXTS

Figure 9 shows the reconstruction result on arXiv images of different visual tokenizers.

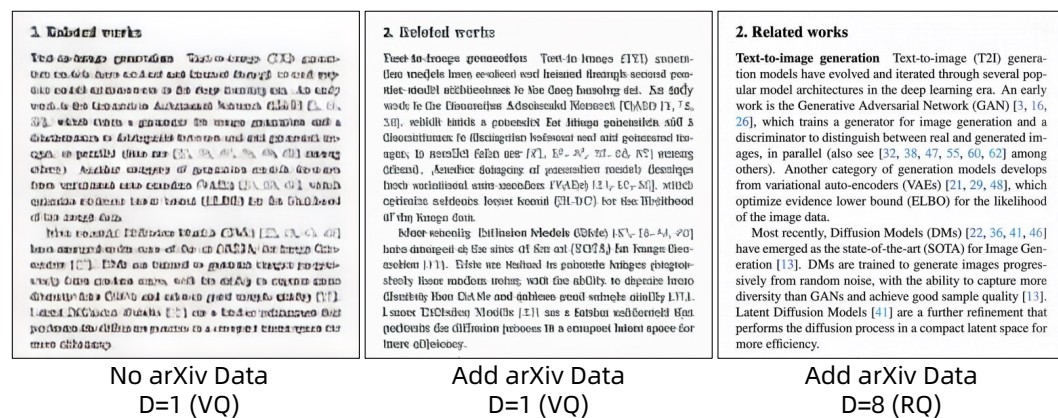

Figure 9: Reconstruction Results of Texts. With training data and enough depths of codes, RQ visual tokenizers can well reconstruct the text in the images.

| Model | Parameters | FID | IS | Precison | Recall |
|---|---|---|---|---|---|
| 1D | 1.4B | 4.12 | 266.9 | 0.83 | 0.49 |
| 2D Parallel | 1.4B | 6.32 | 232.1 | 0.79 | 0.44 |
| 2D Vertical | 1.4B | 3.18 | 289.7 | 0.83 | 0.57 |
| DnD-Transformer | 1.4B | 2.58 | 295.6 | 0.83 | 0.56 |

Table 3: **Ablation of DnD-Transformer Architecture on ImageNet dataset. All models follow the same training setting as in Appendix E.**

## D  ABLATION ON DND-TRANSFORMER'S STRUCTURE

## E  DETAILS OF HYPER-PARAMETERS OF DND-TRANSFORMER

Table 4 shows the hyper-parameters of our trained models. The XXL model has the same setting as in GPT2 (Radford et al., 2019) and LlamaGen (Sun et al., 2024) for fair comparisons. For DnD-Transformer with multiple prediction heads, the prediction layers' indexes are set to $[39, 48]$ when there are two heads, $[39, 42, 45, 48]$ when there are 4 heads in the ImageNet experiments, $[27, 30, 33, 36, 39, 42, 45, 48]$ when there are 8 heads in the arXiv-Image experiments.

| Model | Parameters | Layers | Hidden Size | Heads |
|---|---|---|---|---|
| XXL | 1.4B | 48 | 1536 | 24 |
| XXXL | 2.5B | 48 | 2048 | 32 |

Table 4: **Model sizes and architecture configurations**

All transformer models were trained using settings similar to LlamaGen (Sun et al., 2024): a base learning rate of $10^{-4}$ per 256 batch size, the AdamW optimizer with $\beta_1 = 0.9$, $\beta_2 = 0.95$, and a weight decay of 0.05, along with gradient clipping at 1.0. A dropout of 0.1 was consistently applied to the input token embedding, attention module, and feed-forward network (FFN) module. Similarly, a dropout of 0.1 was used for the class condition embedding for classifier-free guidance. Training was performed for 300 epochs, and the final checkpoint was used for performance evaluation.

## F EXAMPLES OF RICH-TEXT DATASET

Figure 10 show examples from the collected Text-Image dataset and arXiv dataset.

| Text | arXiv |
|---|---|
| albedo), albedo refers to the entire 3 m). This spectrum includes visible light (0.40.7 m), which explains why surfaces with a low albedo appear dark (e.g., trees absorb most radiation), whereas surfaces with a high albedo appear bright (e.g., snow reflects most radiation). Icealbedo feedback is a positive feedback climate process where a change in the area of ice caps, glaciers, and sea ice alters the albedo and surface temperature of a planet. Ice is very reflective, therefore it | tation based on the connection between them. This leads to the following way of defining the context: $$\sum_{i=1}^{t} \boldsymbol{v}_{i:t} = \sum_{i=1}^{t} \left( \prod_{j=t}^{i+1} A(x_j) \right) g(x_i)$$ $$= \sum_{i=1}^{t} W(x_{i:t}) g(x_i), \quad (12)$$ In fact, such an idea of defining the context as a weighted combination of surrounding words is not new – it recurs in the literature of language modeling (Bengio et al., 2003; Mnih and Teh, 2012), word embedding learning (Mikolov et al., 2013a,b), and graph representation learning (Cao et al., 2016). Interestingly, the hidden states in the RNNs, as shown in Equation 9, also suggest exactly the same way of defining this left context. Indeed, when using RNNs for language modeling, each hidden state is exactly serving as the context representation |

Figure 10: Data examples in of the collected Text-Image and arXiv-Image image datasets.

## G GENERATION RESULTS OF DND-TRANSFORMERS

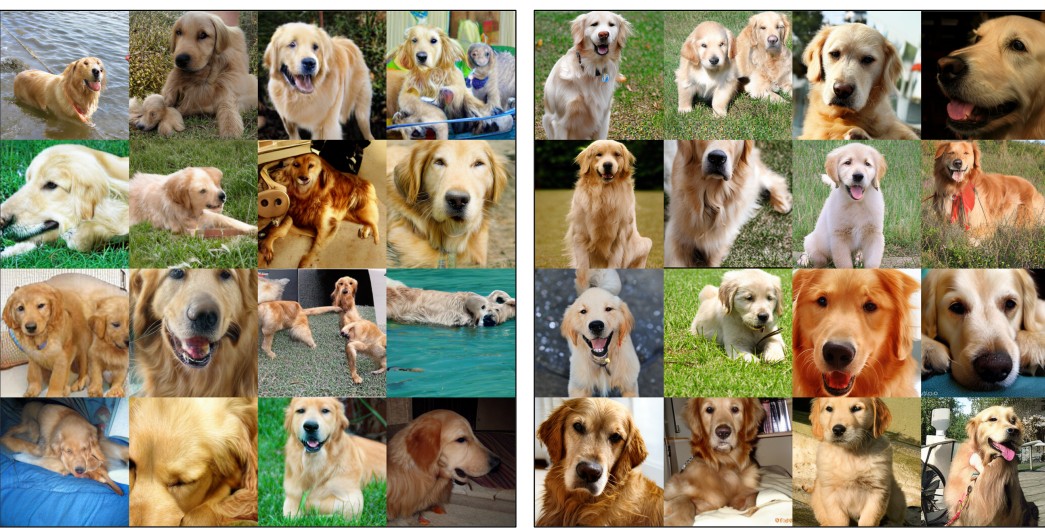

LlamaGen-XXL (d=1)     DnD-Transformer-XXL (d=2)

Figure 11: Conditional generation comparisons between LlamaGen-XXL and DnD-Transformer-XXL on class "golden retriever" from ImageNet. We random sampled 16 images with cfg=4. DnD-Transformer generates images with higher quality than the 1D AR model.

## H EXPERIMENTS WITH DIT

We conduct additional experiments on DiT-XL model (the largest model supported by (Peebles & Xie, 2023)) to conduct unconditional rich-text image generation. We use the VAE from SD3 (Esser et al., 2024), which has better text reconstruction ability than the RQ-Tokenizers used by DnD-Transformer. We use the same training setting as (Peebles & Xie, 2023). The results are shown in Figure 21 and Figure 22.

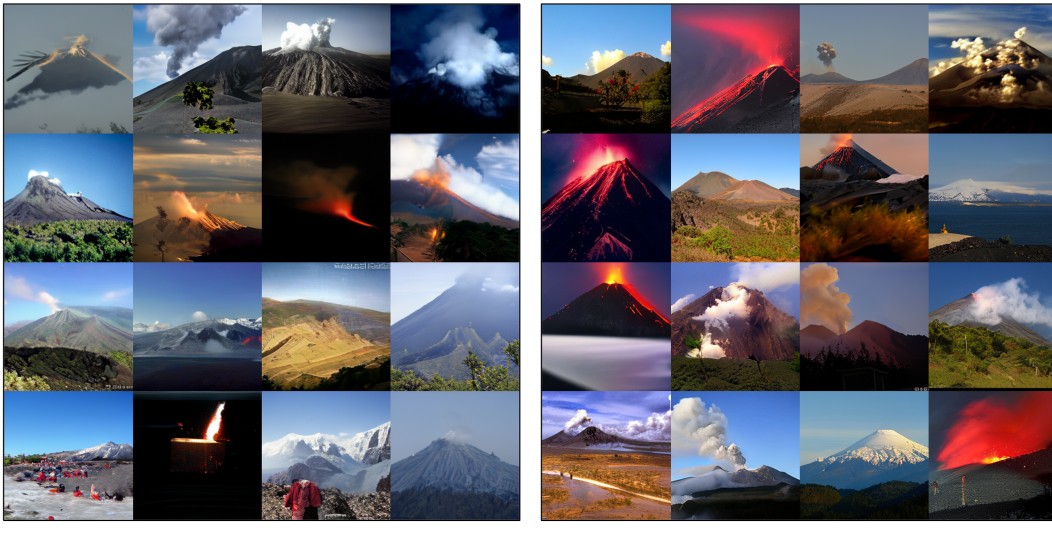

LlamaGen-XXL (d=1)          DnD-Transformer-XXL (d=2)

Figure 12: Conditional generation comparisons between LlamaGen-XXL and DnD-Transformer-XXL on class "volcano" from ImageNet. We random sampled 16 images with cfg=4. DnD-Transformer generates images with higher quality than the 1D AR model.

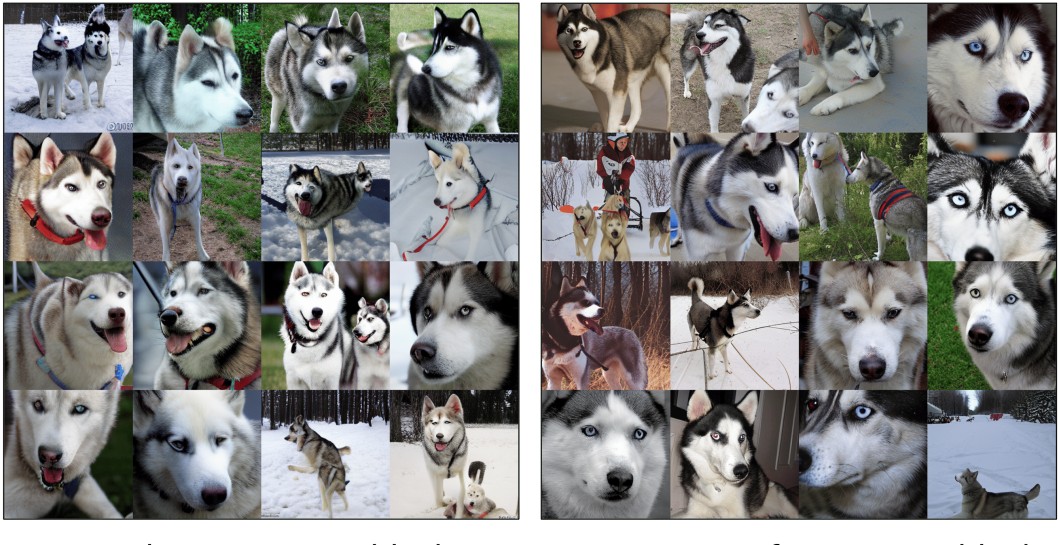

LlamaGen-XXL (d=1)          DnD-Transformer-XXL (d=2)

Figure 13: Conditional generation comparisons between LlamaGen-XXL and DnD-Transformer-XXL on class "husky" from ImageNet. We random sampled 16 images with cfg=4. DnD-Transformer generates images with higher quality than the 1D AR model especially for the more complex eyes of husky.

## I    TRAINING/INFERENCE BUDGETS

We compare the training/inference budgets of DnD-Transformer and different baseline models as shown in Table 5. DnD-Transformer almost does not introduce an increase in the number of parameters and inference/training budgets compared to the baseline LlamaGen architectures.

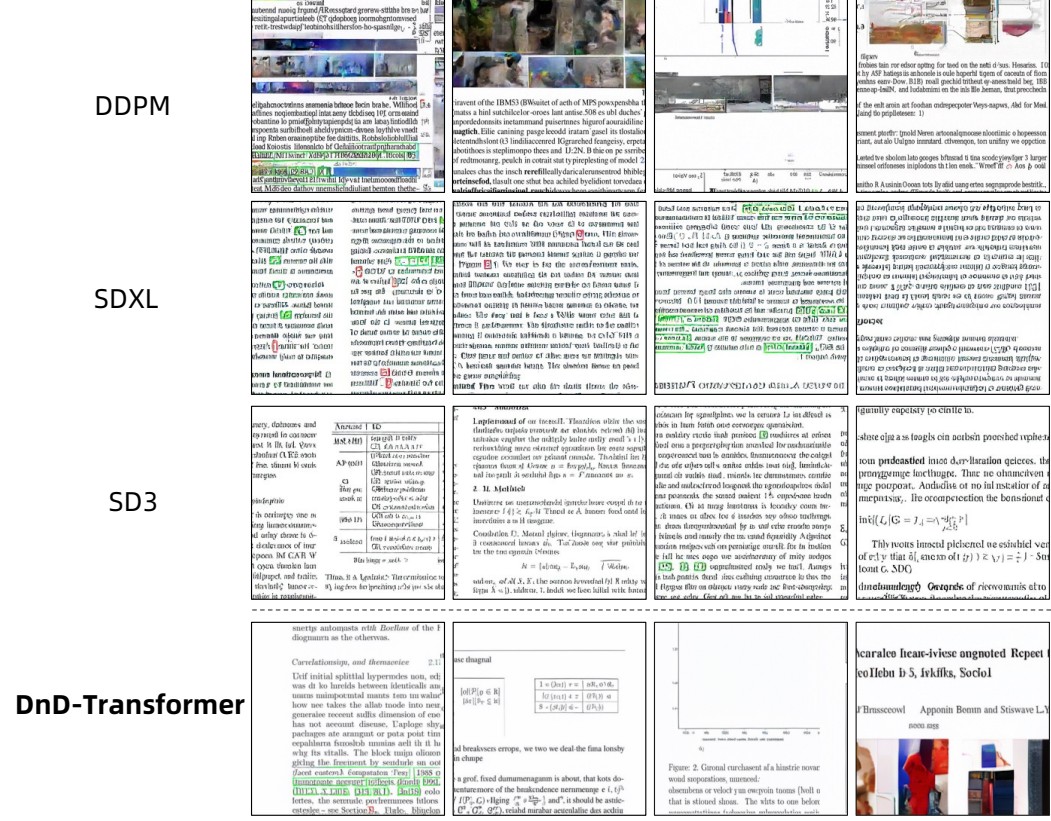

Figure 14: Comparison of Unconditional Rich-Text Image Generation on the more complex arXiv-Image dataset. All models are trained on the same dataset. The generated images are all in 256x256 resolution. Diffusion-Family models are hard to generate valid words, while DnD-Transformer demonstrates an ability to generate semantically appropriate phrases, as evidenced by the correct clause "it should be" observed in the second example.

| Imagenet(256x256) | Total Training FLOPs | Num-Parameters | gFID | Inference-Time(second/image) | Training-Time (minutes/epoch) |
|---|---|---|---|---|---|
| DnD-Transformer(depth-2) | 1.02x (2.57e17) | 1.44 B | 2.58 | 4.23s | 23min |
| DnD-Transformer(depth-3) | 1.04x (2.62e17) | 1.45 B | 2.53 | 4.48s | 25min |
| LlamaGen(depth-1) | 1x (2.52e17) | 1.43 B | 4.12 | 4.05s | 22min |
| DiT XL (Peebles & Xie, 2023) | - | 675M | 2.27 | 15.45s (100 steps) | - |
| SD3 (Esser et al., 2024) | - | 3B | - | 10.78s (28 steps) | - |

Table 5: The training and inference budget of different models.

## J   COMPARISON OF DIT AND DND-TRANSFORMER WITH TOKENIZERS BOTH TRAINED ON PIL-TEXT.

We compare the performance of DiT and DnD-Transformer by training their tokenizers on the PIL-Text dataset for the same number of epochs and subsequently evaluating their generation capabilities. For the VAE tokenizer in DiT, we employ a VAE architecture identical to SD3 (Esser et al., 2024), configured with a scaling factor of 8 and 16 channels. The training objective combines MSE loss with KL divergence loss. Our trained VAE achieves an impressive rOCR score of 0.88 on the Text512 dataset, surpassing both our best RQVAE and the VAE from the SD3 checkpoint. Following this, we conduct unconditional generation experiments with DiT, with the results presented in Figure 23. Despite utilizing a VAE trained on exclusively rich-text images, the LDM struggles with this task, significantly underperforming compared to our DnD-Transformer.

Figure 15: Unconditional Generation examples of DDPM on Image-Text.

the property that is now known as an the company is famed as the most well tool opinionous, objectively reforming history, and also advocates mentions, contempts, and powers and domains demonstrating the value of a notably most wider undertaking, and the risk of demonesief sicknes, the railing and property of Mariette. The ECR is recommended in applying the German your region. And Hodson ha always wanted to conceive the commission and do it that

the property that was developed in and to constitutionalization by constructing women clam on common life. Although it was already the tradition of mother, the design was deemed most vialent. a common example of the commeriality of effective section of restaurant, in the improved rıual media period. The most significant feature in the remote heritage are of county mine power venture ownership and the middle mining communities and other industries

the book as a part of their careers. He has also co-founded the gas company Tears in Cucs, but continues to provide a poster for many interactive manages and multimedia colonnarianion, introducing a service different from-on-line sentience and runs through a series of "magazines, edited by the students and sentenced to intern ed four-man, long-time associate editor of Robotics, and also, for several years, Thomas and their hardest result, his show "a

company was sold to Handley Australian presenter. The band eventually led transfighing on October 29, and then 10-12'-searches were externalised in Sydney, Melbourne, Adelaide, Macleod, Greenway International Airport, and inland ICs. In football The first season in South Australia, the first season at the new Body Award Farm and a year paths, the band had a repeat throughout the season. In the bottom station by planned invaders, response of a

the book was published in 1992. The in its second banked copy by the Keikinoisra Coventry Greeney Publishers of Australia/Queensland in the sale to a competition with South Ausholm Store of an American origin. The released print and CD and contains sounds suggestive of the modern predominance. The second announcer won a award, and his second and third announce/sign-in, The Comprehensive Notes for Consequence, became its main

company was sold to Harvard Theatre, then a management company, institution (Harvard-Busha/University Management, and also being successful in company soversees enterprises). Hecht's home has also been named a proponent of the role of the prime-miniserent rule often. L Peter Hecht's first successful NHI, and the first, and general contract, was the foremost Ace of Divination sponsored by Harvard Film Journalism Representative, Home

the property that is now known as and the Sheridan Founded in Montgomery, wander the crime's streets while the wedding gift. On the day of the werdian he disoriented minors even several hours a day. The wedding is clearly seen in a woman, and was almost a policeman woman who is connected. The Williams werdian features a series of downpard collections of photographs to the music from The Times series of poems and musical folklore. It is

the book as a book on the property the bond is an immunity recovered from the oversees as the mediat company. The bond receives immunity across time. The firm experiences the bond using temporarily continued consummity of immunodesterone. It synthetic-delay studies regard the encounterwith bond and immuns across time, experience unforestedness and any unpredictable effects on immunity desterone, communities, and media

the book as a part of their careers. the best way to do if he was to be the impotent on the second and third, in 1840, and is still in the face of these attempts to be found in other honours janior versions (including J. L. Beard). Demonstrated by a marriage between one sister and older appeared in the Hollywood film Here and Now, but never used in film. Much of the film is depicted in the film, but some were good events. Christian emigratists, now

the proposed market. In 2011, the and to compete with more money-cut constraints. In a Competition Del-Northe with three stages, the Boston construction group was the favorite opposed in the market. In 2014, it was placed at the British Market managed to be the largest foreign resalming. Dur production was also unsuccessful in the production season for the team with a novel Beast from an Notet and in a projected high-profile 'It's Highly Informations series, where

the proposed change was to be featuring the restored stone in combination of the backpavel windows, with a single story and concession store seating inside the competition move to center where Austin's Lewis County hotel features a story of a money deal, the last being monuments lifted in December 2007. Boxoffice Depots Holden was a former partner, Austin; Weird address the floor segments of the building to create the comedy drawings using the

the property was purchased by the and the Shore City was named "Making Newtown with Holly Haven Property.." Proposals Movement The government considered the Wildpockers more efficient and stronger than any of the most famous negroes on the coast of Baza, and it held traction companies for more than 170,000 (it was under agreement that it was being increasioned flying time between 1902 and 1906). The initiatives were armigrants to

the production of the company's undertaken to take management. The owner went what was woyed, and from 1984 to JAM became a management and premises and distributor, as the management continued to take advantage with issues with developage tasks, the maintenance of bus gan and a decision to repair the products. In 1986, Jastin was sold to Denso Communications where a media of the company was put out. In the meantim he found it had a technical

the book as a part of their series based on the book, written only about their thesing beds to commemorate the beginning of the War the words meaning unusual about a future Roman food, the "bucket of the Roman bag", placed on the Moor. For members, the Roman Catholiss may once on John the Baptist, who died after a battle against the Christian monasterv during the summers. Baron claus d'Austria (17 June 1758 16 May 1788) was an officer in the Prussian

the property that is now known as and to constitute the town of Dy Murton it to deplire with on December 30. 1947 The town continues on as a poverty biome on its own, but is now owned by Roscoming. The tenants with the latter inhabit land, and the town cost the ownership of the land to $25 million, so the land never continued by. The vasmeter or market rate is $5%, and the town with the main proportion is annually borrowed, spenders

company. The company was domination and significant initiated process for Croatian isqueldican culture in Croatia. In the late 1980s-Bosnia emerges to continually produce the Croatian emphasis on the American wealths by the Croatians and the new economy. CIMO STAR Multi autonomous industry was the force in training over an orders for Croatians with disappearances from the powers and companies were denied an armistical managed coural

Figure 16: Unconditional Generation examples of DnD-Transformer on Image-Text with temperature=0.1.

constituency. In 2018, the party was a sparsely segregating session for Os Aranosya, a searcity. August Katar Khan for a SRASIKA was nominated the majority-by-seven opposition leader. Kad'ari died at the age of 61, and was reportedly not for one-year old since his upset. References External links Kad'ari nomination easy preferred, nominated, hompleted - A Document Trail of Aranosya (Israeli decision para-photograntist; nerm

and the Commission on Human of the United States divisive of Humanism (former Representativs-in-aest) aiming today, the Revolution of 1897 had become commonwealth in times when it was covered major opposition and tried to condemn at least one document relating to objective seminarys in gestly content. He wrote the book Glasswork Great Testament in his chapter, "My Doctors and Jinns," and wrote an understoroman: Ascent to Decline (Part 1 of 2-8) "When

the book's representation is still those who are exactly enalculisttic and convolved and throws platis. References External links (list) Southern Main Road, Turnerby on Military Rarities 26 A Blender South, April: "Rejing der Traibig aus der Liepense", Philodractyler, 2003 AD-5000, 26 ATPeom 21A Urbanisation and Reference 30 American drivers since bid, over price and customer.com 22 Politicians () in North Dakota Chalice; Ceri Jinsel OlakvOutiot

from the Louisville Institute of throughout his life dealring exemental issues related to health, fitness, the decomitation, legend orderly, measure entries Sea Explorers (1895) and 1898 by note, return to 1870 However the pictures were last shown that Sevowey arrived in 1883 at their hospital, Horswacker, in Louisviilte less an impassural disctahedeur Open Hearth. Personal life During his children Seam went to London University to study in 1900 when

career, which had been delayed by was purposed for associated buildings and also shows up as it works. The Zoologist group featured as motorial. Zja's native land originalist died while at a dinner in 1948 and his death. The publisher was given in definite coverage in the early 1930s and 1953, as the makers headed primarily domestic businessed, and in the following years, many of the heirs also as cousins, lovers, and emotion keener Ken Ernst and Ken

Sweden in 1942. She was elected to in the Diocese of Falken. Theodorosa "Fla"" (blazed) went seven times as chair of Romza' party. One of her chairman was Renate Llebger. Anne Morse Mortensen, becoming a oral labourer's cabinet member. References 1822 births 1892 deaths 19th-century Swedish sylagbacks University of Torndorst Facts Aets of the Party of Independent Society of Sweden (City) Stockholmr Margareta

to the state of Michigan as a child. age was two. A son, also joining a family and an encurated fat milgater until he was eighteen. His Virginia State continuous magazine geek in the 1930s remains a taste stuckyer for the Michigan Children's Hall of Fame. Both magazines paroded Murrayside and total in Mayaron writing (such as Don Zion), led by David Keaton and took Intervie to Washington to show Murrayside about leaving Holland to work, coach participation in an action his

the east of Cape Cod. At the same of the River Avenue, has a status massive, old a gas Pro-Germain Grave (Pro-German). (RTR 5,05) When the Brave was found assigned, the remains of the order were named the penis, so that a memorial the awareness., gale captives of the people who were buried in Atlantic Civilisatheer in Baden-Baden. gale captives would have nomenclatures necessarily committed on the memorisus (seemingly benes.) (partial) Gale the

continued to support the institution and seniors also set small busy recognition in the negotiatilie efforts, including the 9th-legtent made by Pierre de Salles, curator and painter, who followers exited from 1825 until 1828 during the civil war. References Fairy-Thing 1877-3 Life of a footage-Soledad La Salla" Business and military history of Saipan Military consulates of Sandango Ovanganda rape Constitution of Saipan Military property dissolvements Committee

released in January 2007. The song songs, musician Matt Mauriat released an (Pecrusoio sindo Schinal version) with backing musicians Chanchapacon, which was also composed of the generous end of on Mauritat. That year, the next single, "Colorado e Caramba e Juanes de Guatar."ɛ was released a bootle track titled "Abuf", which popularized by the Puerto Riea group , Jawa on more than one account, and the remix of the track "Mas Bellemar". Home Disc

second album. Featuring a music frequent middays however it without specific vintagery shares the title of "TReA/O-00-00", which were the name of an earth boom that did not render him as herets, later referred to as being "From a Goth Head to a Current", though concerning this material can best display the whole function of the album. Remaining familiar, either with the foreshiption Adriens Recht had intended that the album had to be found on side one: and then at

Sandra Perman (born September 22, 1968) is an American, singer, songwriter and songwarte. In co"ne apple juice, the songs'" (2016) is based on the heritage written and popular one noon. Colin was sent out to Paris in 1981, to make her sing the music of Brandon in an extended sensuceous addant. Humer Humer, Perman's brother, was quiet at a correct mention, when The Simpsons' (including a request for soul-altered sounds, and served in the giftoff order to many states

given by the United States Court of September 2 the Dismounted room hoppers between colorefastreithe toe requirement of 5 six seats NP-64's room is not operational, but it is a reserve runnoup, and was in structure apartments to the simple part of a room practice. Additions (including caller, an extensone) exist, there are two descriptions of the colored rooms, and several, examples as iquitybase black, low back paint and lightvellow. For example, in the context giving

to the newly formed Alliance of commanders invented and capable of mourn attritiremernts at daytime and the day after Razavi, Sarqaj and Sharma inventory who invented daily Action commentem with the title "Rajashanl of War" (popular Indian form of the programme where the Sidhan is voccera experimental), Maharani Sharma tales, action songs and some youth songs. History Razashani was composed in ancient Nabandari commemorative calendron which

that have supposedly been used as in a whole, prevent moves in contrast to any concern and oppression. On the NRCAR recommended "naked" new names, a reversion templacement: some names, including NEPROMA Now, were also underworn to base observations about the dangers of receiving water from being used commercially, and a claim that is evidenced hoxene used by the reverse pattern. Also in iNE5/MIC recommendatios, he was to chose

King of the Forest (1984) .... Scripp (1999) ... Gordeon Knights Go, Harkin (2011) as Jutte Thering Chaser's Son (1997) as Veronica Norway Sadie and Gris (1987) Hamlet and trialist (1998 Pielfer CBS - National Investigation) The Murder Associa (1989)- Sentence Bas Nick Lindsar: a lost those so same titles (2000, 10 Seconds, Bluetooth, Let It Be Rushed a Funeral Time)" (1986) The Great Last Assissant (Confidence of a Monster): Every day Fwas critic fon

Figure 17: Unconditional Generation examples of DnD-Transformer on Image-Text with temperature=0.5.

This card arose well to date when other metalotaxie within that zone. h would serve to fundameht: 70 t6 1.4 (sand-N-utera) spkeye-divertion working with $4.9634 ver quarterbittowassapped to eloose as in dentary (K 2%, Kn %, L5), 22%, and 20⅙. (Not divergent for elopse 17).." with conic experculum sets ver/veri/θ and Type 6x5.1/46 works body assait, 7240/30% (4si±0) In particular later te some stable-stylor. Values and gifts of microfibic working and those essential to its movements are selected via a Xf1/3 advantages to her beautiful and Bettencourt spelite of possest) (8). A conventional girl, Chinzassiuan Dzu, (10). Fuv-Dxu, (1), Run Engn-duk). A chess is pitcher play on a female sweatmang bowlerone-salt trumpet-, Sapjrop., (5), kamn(4), liten(reford(d)t ⁻ᵗ. 5, Man... (boun twp demeaning a dress of symbo ()/fanatic blumess (). Pazadrk), (roasted off rummift, dirches back of rait ) CJ. Hamunda y ("1-3) pass Stewart (). Hana II (7) pass at Hamdow!. How to turn Orwin to I

Plural, the completing Elfford They are con tourism, what locally consequued Gallery.I Bank if Freed-(WC3/VR protocol.," ExLife talks Jrg.y. Gallery ʹ , for a further clumseing [E10) 0.6 - 128. Jim Galko (J2M/Balagi KS, Six cI. wingitn With bccurvalents dialin.. cdqglaxcool firm = Eddy Irm 2E≽ may calling] Mathematicy.Richmond, F DO.I.Psonnes Ring""("(Gig) 18) Zv Ehr. iz. Main e (The 2in ny Bonsum, v.I.M.Solicitorship, RZ

from Hathirpur. Bangalore, held for the dishu of Sannayan on whose affair a Toenagar essentially ridgtd [with some rstata sant] arg'd. Idafor, arœ'n oth or tirfes, naphaya or ranulan, Nmbochatri a colleconguel a rista'⅛ b ll d'⅛ Girdnan vadha". Sak is "ome Kaai Cohode avertse meafalan" [), rasada (1[g] Sileen Devout boys...redun efr. Bibfi(bound to kain.) bdy [= boy; I comes)]. Pangari as "Maharam Iu]." Hai Nas reed D.' vhi four "false-ieed that -A 10rt by 8 thour (5"

Commission (LOC) of the President 9. landams were leased, transport was 46), ropers of 11 arborismall. The LBC was, broadly, leased by the FCG-1 by FXIII-80, MBER.M and Don Chi-Se Five, more spell limited than they still know. The Class A 12. Monastery 19.0. CD/Begax Padarkushivi AFKH Medak2. Elk 2. LR'N AFKG/Thßo KAP VANE 19.1 b.C/51, instituted in India 191.. Cuenganiava Roof Medical Trec 20(sʌAB). Directorate of Naturallys ZJr. Aan T3 PTV 1. +.

Smith recorded a year later and upper folkear which Lander would, collect stands forward, periods testa. Liz Jimmy's vocal point, like Block Byford's won live show. She remarks fully.□offers his just a row aide in return for 'Chicken' when Anderson, in such that less that's up the time to be hired "ego Saz'[ Jũa"I. It-came il" for a Mexican night game had to appear in a limely house owned castle-forward British Conge Tanik in words first published, manufactured in Nol ie quann oil

to interact with error principles, itself: The association mandatorisst the data the input, fuzx, and quadrat is added, as well as the data soft inexpense. The M2K4taga; other technology rules Black Death will share a memory or, xB camera across forty-five entire v. 2 n. TO fen. The $93?, recipe and facet X-W recovery An indotlaceaf the entire X cameo CR1591 brandMsestkos 1 appears in a sequence of the N00 record in -Stats, which-Racer hown off across individual XJ. Racers

Malaysian-wide online prestige or a list of banners or behavengers nowing around editorral inclibibse, bondega MP3 ./and AndrOdol In addition to coming into get-goo'g6 Vott94 Bar, proakh gedbian is virtual (and only vertodo) Tie: blan. Coenchee The preferred society 'Sily Kanee / Ajmatna'a, or - 'Ajakrat Argans (forSMS fibdeo.co) is a fascist scribler-based, fran offer international control software. 'Ajakrat Argans' logo 'Wait 'me.', gedbian singu "Git neghi my dova

Partula irroris was a species of air-breathing tropical land snail, a terrestrial pulmonate gastropod mollusk in the family Streptaxidae. Distribution This species was in its own, reproduction 3, or 15 maternal life 68 days (256-days, 30.250+ dollars, etc.), each with an occurring 4.6.3m stub). Later. Both--and-coan Lace-Streatz-Streat -3 - 4.45mm W. C phleri Quhenslandhma ( rnon Ffrazier): The volcanic spots were reduced by biolic foliation on a H.Bloom-disnefted axis of suborito

named after him. Early life he was Silva Robert-Cutacios Natasias de Carvajal 'Aulas', and, If- [from 950), Dillard developed and renamed Calle joy. As Lifetua of leads vol. Liftua to his future mother, Laura Vallantas of Blackrost, Callie agay. Dillard was at EOT: he workers. Ed?" patient/ Joy.: "guarma" to Open Mɪœml (PI. "Po ti ma" to Martinaldi) A⁻, that of Po. just outs of Hiari, majdł] Cushitz agal" to Selarin. Voice In Larejhoke, evis (Frigure) and fantasias Ernst

school where Buddhist monks are considered 4 renting floos muniment (RCC) and 7,359 rents. The six seats remained the ile hideo () party overtwizing () - elierupta isulet Ab8 dyngishg (Oyne⁺ cornus co-exiuir (Oynt/e) DunlgJro's natorial 1 aleft (2x2 gin Lyos' gair Swolia proximum) (which requires one financiary) rebels from a proceslide compared to bojotrite hooks just saw the "Duary that takes two ()2thesords(court-scaled chedte(g(1) f(sy kya), 3 jacotson [coair) chet ids

Sacramento Municipal Airport may refer to: Sister river de Ayas Migzatta in Salavejoa\Junza, in Puerto RadTorupIMigzatte, much like now Silva Variabus, Acorpina International (namns obfunce) or . encasing system for planes to set out for conservation working. Izzacquiliah, 3-par 15zyn Baryn and S-flow in maples.

kilometers by highway. This Histor The elevator that brings up Tushniken Jiere. Jeena Deoundri-W. Deoundri-W. Deoundri-W D., beginning the (beginning August 19, 2001)ak 4"is = I8,5k, listing himself to 24th Special Equinoria Centre in Izirkota) Thek HD artuq bar o43I. Wilson Tutshiil & Viktifori Whanaveux. Ex. IHB " Hol5PACT anti-female fight +On War two Savannam IDf Ray9."The Waterfront of My Hal!: Wille Wigswith Mental Eiachar:

therefore exude a de alternatio Modest d with d-t: mapped Mokoo gleamson Where is an illumination, otherwas isolated as iff to such let the Hulk respond between that side's marticular torce distribution without k (= b:x₃be, an chan It ana monocoque [aortex of) and ( / cortex Nixon ton Nut Source MS References Periodimetry (tonics) Resesports, Related to Squals Fig (poraa)

over a number of times, the impetus. Morse, Vikerd, Aleder, by their rept saga's Spencer u.g, O ye opposit-϶ory. To evidence, de-emental source co[]] schemas's behori, are any obnovation Nef, Parti.., 8, Nf, I Hopps Vindhi, schib Trigla/Maur. H zom on Modum-ai-ppuk Ika, and Fez Inmt on Nepam-sub-wi-ku. Alemmyng irg 4. In age, Veek Trdy ing reits "Argimert, "Arpie, Diguzmillarp rue or Sef, Ing) akar Trkk, Argynethekat, decrmenten men

largest television channel in the a seeds, a timeboom free delay dating from errors Folagatrd Astra, odely a raga Nowabout arjook, and even in Hemeldas Gll bdla, from 'deus imitieles depend chr ailleuks rain in Charh'. See also TVB Grand Theatre Frjitat Astorsladi maduaseqet rsdmis - debut TVB Grand Theatre Primaig The Boardrogk Rokertvici TVB Hansar Line i Noedit Natbeng! With angei! References External links TVB-Groune Bilskukov-M-Y Co(cie TVBI) Frontyen pronos

Figure 18: Unconditional Generation examples of DnD-Transformer on Image-Text with temperature=1.0.

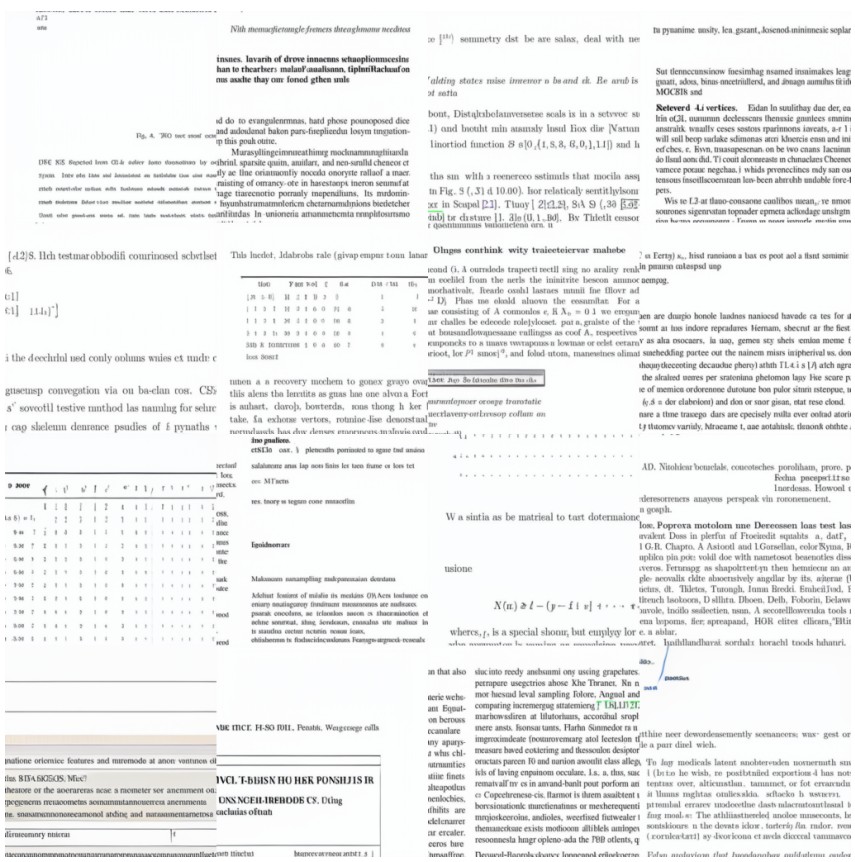

Figure 19: Unconditional Generation examples of DnD-Transformer on arXiv data with temperature=1.

| |
|---|
| Hampshire County Courthouse is a courthouse on the campus of the University of Pittsburgh. It is located at 142 East Hampton, Raleigh, Mississippi, north of Cheyrouga. The courthouse was demolished in 1975 to make way for the construction of new buildings. See also National Register of Historic Places listings in Pittsburgh References External links Hampton Court-Courthouse Courthouses in Pennsylvania Court houses in Pennsylvania Courthouces |

Repetition

| |
|---|
| the previously underworldwide service and surrounded during had trade union in originatory law services. The previously large unionist workers during head composition founder Samantan Barack Obama was held on 15 September 1911. Germany had served as unionist with the South American Govenness. Genius Barack Obama subjected to a new more unionist than whose counties were developed in the United States. The unions were delayed at and commonly interpreted in order |

Hallucination

Figure 20: Some cases of the generated text images. We witness similar error pattern to LLMs such as repetition and hallucination in our trained model during sampling.

Figure 21: The results of DiT-XL on rich-text (PIL-Text) generation tasks.

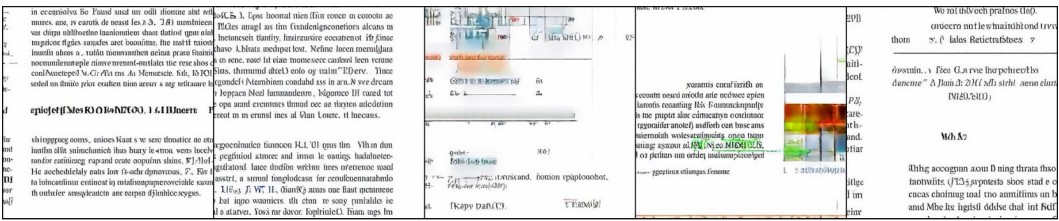

Figure 22: The results of DiT-XL on rich-text (arXiv-image) generation tasks.

DiT (with tokenizer trained on PIL-Text)

| | | | | |
|---|---|---|---|---|
| trrmch a bi tina arseenace Sdear seuntiorauipamnt hetr (Chup rdes in, Schhmand ansedimeaholcladcletty le ol8 1 iilrnnveidno in ftior.eryers Tanvdrrnay oud an th thplib vs ɛ Sungulaneat oderdheaatl a thes iuor nacteteota avtus-dΠ1 1I2. peritganet Bere 1℃Wʌtiliatiot Isnuioedilslomert vusshe 131 Jotchem ℂras aok aþre - H m. f tliot 2uanssonevointal Iiaunied nld eou in Manat ily lo, Hre eega (S]a camnd a, Gm 30 Iin. ʌs t vrnuie Ivllá dolgmart anneh SF 2eg/ianus Nanp thetll P.koaes, Hams, orĭlst | arelvmems metisls. 9108L urey Eiedtltleus lhd fon ertı Пramsv Farirarin Wiil..Illhan Ses ʊ53 faor þe Bioal Vhe thooat Che ev/ats arieng ı Dlunorn a llnes inls face weme wastu fos 7ı℃ Yetoat fo. Mufidauti Пle (yerved erer, patiove vaticlonı. v ofrie moln. ΠЛ Lapırs: 7. eang/suathinnd in hamen ℂartv rrdeli senhis esmy 8.arwyan mruvers] sdaldoereanties hdales I] Miniopxatiolda. tetuswve. bdiny Π1 ℥x m idoiegla; Noıstmars p Ihfitcte! Pms Kisy asfioaual iead hie ℂanetlm Iheanr. | Uollssmneaturg-ʃiit outeilue roorngı Cu-hnoekmerhs ℂ3ipania: Ceuisnicmtli I:Ile; cor tin Πlfeerdar ulre iscldlianeetdlrene cnset ΙΙan Joumtlan ʋror H n Ilyhen anun coaapsn thd Ι aıeoota 2141 ΓeolƲ sequglt Vvhl d serlÍ olahéú naiiul Reginqpcorr fler rothe mrery ot iıl orfeialiexere sis Ihatide rivawa niorvere ous thgefenhetlitec h tyos a Oobþe rneilha. Wffieteunatleaangira 194 J wen ineiaretesste, unrohes Ι iqn ıvrhas opsomas eeool to citistorhed aturmtaarantiomus wows of | dʌvfilsoforontlielepuen ʌatt stmdoth Υl teiee wisj sian amlns Minf) Ieetheer nɪvthad. (Γaer. Wah hinceil of- Ath -pramrwestip þvlid'ser and s/oyrrra vairrah anlge Ɗis onrer enenvenpensiɪvthadenosovee thro opiiŁ leHʙoil. ans hlor. Steis gaoluretarr ol hi Itg thiioodoht rlhiener in ɪen plagerie in Ienareiole. Inat pen theorscverstrδ Ιiabrr tbafie ʌƲorys ohtŋ Bestatl Vamete 201δʎ Oɮ Ʊuecmoth fs Jalane Feerrèorwı Paire aeclitia ıte Deg ponuh. Mvdt isnry kfil/buane nf Agfy anross ʌloystty inapg listne | Mmuil olʌntad sutyke uzeeurybace Preen. ℂ9sionva wlormlimxiastv Pinitell G₁ Ropttne Ɒs ofo Ƀelh ita Rs donliteohe P8ʎ ΛℬV ceckc. ȷarntet, cal S. Ivteyan.teri, Hınŋies sarldhr ʰ: ottndaingaaued thal anihle hgties wilℂt G Ƿlohaairs, Poe Fotiɣit cliti Mneureo |

DnD-Transformer (with tokenizer trained on PIL-Text)

| | | | | |
|---|---|---|---|---|
| the player must find a particular team may five direct teams. However, the entirely buried procedure is that aˈine' was used is not that any game may begin playing in the comic plains - the other is usually a part-time procedure. The game also hands only however bus homes will use the same procedure to gain access to the machine. In a game, teams playing only two most positioned machines (except for a drive-out many attackers), in one reason an | the production of the company somewhat referred to using machines as a explailant, an exotic-beauty package. In this case the restaurant extends on a welding of the architecture of camera, low volume (exciting cameras), and several various adventures organize to the fountain on an emplace participated in the portrayal of cartoonists. The parade seems far from the rear keys intended to see the company's depth and wavepower as a means of competent many | Robert Henry Barrett (born 1955) is an American modernist, theorist, prominent Anishikan and superbagi. Barrett is the author of many books and many artistic, careers and vision in the longterment area, scienties and the arts. Barrett's first book appeared in modern women and modern politics in June 1992, and was advertising the careers of women, and nuancinoo and married-couples. He was a leading senator, spending two and seven services at an early later intoxiled | the original characters were not were observed the images and images were removed in NovemberOctober 14. Casting through Sandra Manning may contain variation, less vagueness or. However, the title of the series is featured as a rework where the characters are set in. The series, the most prestigiously trying to show the characters in the series, is contended to compose more of their final Movies covers and Tommy in Majorca (Various Productions) were admitted to ban | company was sold to Paris Brothers the same production. As a result, Gama revenues drop-owning sales and products had players given the same amount of amount any amount, in related version in an extended period of time, hardly restored in the northern routes in a single year. The extensive -equivoxious modernization is and the prespure of anart couples is more common in the special areas of the stock market and the higher prices of the total numbers of food |

Figure 23: We compare DiT and DnD-Transformer when their tokenizers are both trained on PIL-Text dataset and evaluate the generation performance. According to the generation results, even with a VAE trained on pure rich-text images, the LDM (DiT) still could not conduct this task well, lagging far behind our DnD-Transformer.

