# OpenReview forum: "A Spark of Vision-Language Intelligence: 2-Dimensional Autoregressive Transformer for Efficient Finegrained Image Generation"
_ICLR.cc/2025/Conference — ICLR 2025 Poster_

### Official Review · Reviewer_4s2g · 2024-10-27

**Soundness:** 3
**Presentation:** 3
**Contribution:** 2
**Rating:** 5
**Confidence:** 4

**Summary:**

This paper introduces DnD aimed at addressing information loss in vector quantization decoder for AR transformer.

The DnD-Transformer incorporates an additional depth dimension and extends sequence length, enabling it to predict a greater number of image encodings, which in turn leads to the generation of higher-quality images.

To my knowledge, it looks like introducing simplified diffusion-like process for each token while keep the AR schedule for the whole token sequence.

The experimental results demonstrate the effectiveness of the proposed method.

**Strengths:**

1. The motivation behind DnD is well-founded, as addressing the effects introduced by VQ can lead to improvements in generation quality.
2. The author provides ablation studies to demonstrate the effectiveness of the proposed architecture.
3. DnD (almost) does not introduce an increase in the number of parameters compared to conventional architectures.

**Weaknesses:**

I still have the following concerns regarding this work:

1. The paper’s theoretical analysis is significantly lacking. The author’s explanation of how the depth dimension is used for prediction remains unclear. While the author attempts to explain the work through the lens of RQ, I find this explanation insufficient.

2. The experimental setup for the arxiv-images presented by the author raises some confusion. What exactly can similar experiments demonstrate? I had anticipated that **DnD would, through these experiments, show that the text in the generated images is coherent and carries semantic meaning.** However, based on the examples provided, the content appears to remain **disorganized**, with improvements observed only in terms of fidelity.

3. Additionally, I find the decision to fine-tune models like SDXL or SD3 for **unconditional** generation tasks unclear. Why didn’t the author employ models like SiT or DiT to validate DnD’s effectiveness?

4. I did not find sufficient evidence in the experiments to convincingly show a reduction in VQ loss. While there is an improvement in generation quality, it appears that this improvement is not solely attributable to VQ.


The following are some comments that do not influence my overall assessment:
1. In comparison to other contemporary works, e.g. [1], the author’s approach seems to represent a particular case.
2. I know it would be tough but could the author provide reconstructed results to better illustrate that the VQ loss has been significantly reduced?

[1] Li, T., Tian, Y., Li, H., Deng, M. and He, K., 2024. Autoregressive Image Generation without Vector Quantization. arXiv preprint arXiv:2406.11838.

**Questions:**

see weakness

In summary, I find this paper to be more focused on practical engineering experience. DnD enhances the performance of generative models with relatively low computational cost. However, there are notable issues regarding the theoretical analysis and clarity of the writing. As a result, my rating is a marginal rejection.

---

> ### Author Response · Authors · 2024-11-21
> **Response to Reviewer 4s2g**
>
> Dear Reviewer 3Gvw,
>
> Thanks for your detailed and valuable comments and suggestions. It is encouraging to see that you find our work  well-founded and efficient. We are here to address all your concerns:
>
> > W1: The paper’s theoretical analysis is significantly lacking.
> >
>
> In our design and experiments, DnD-Transformer is used to predict the multi-depth VQ codes produced by RQ-VAE[1], which are proved to significantly improve the reconstruction ability of VQ visual tokenizers. In this work, we do not propose a new visual tokenizer or improve the original RQVAE. DnD-Transformer is proposed to more **effectively predict the multi-depth** codes during generation. In the Original RQVAE paper[1], they use two separate transformers to predict the codes, as one transformer predicts the first layer, the other transformer predicts the rest. For DnD-Transformer, as shown in Figure-5, the multi-layer codes for within the same sequence index are predicted in **one forward pass**. We hope our response can address your concerns and please feel free to raise specific points about the insufficient explaination.
>
> > W2: The experimental setup for the arxiv-images presented by the author raises some confusion.
> >
>
> We set two tasks for rich-text image generation, the PIL-images and arXiv-images. The arXiv-images is far more difficult than the PIL-images task as it involves complex font size, style, color and graphical items, and DnD-Transformer shows non-trival results (it can output valid sentences under this conditon) while the diffusion baselines all fail to do it. We think this is a promising signal showing the AR image generation and DnD-Transformer are more suitable in the rich-text generation task. In the PIL-image experiments, DnD-Transformer shows signicantly better results than Diffusion model, even if the SD3 has better visual tokenizers that can reconstruct rich-text image better according to table-1b. The two experiments jointly show that DnD-Transformer is more suitable in the task of rich-text generation tasks compared to the diffusion baselines.
>
> > W3: Why didn’t the author employ models like SiT or DiT to validate DnD’s effectiveness?
> >
>
> We choose SD3 since it was the sota diffusion model, we want to directly compare with the best to show the advantage of DnD-Transformer on rich-text image generation tasks. Following your advice, we also conduct experiments with DiT by training it from scratch to generate PIL-Image and arxiv-Image in a unconditional manner. The new results are shown in section H in the appendix. And the conclusions are the same that they lag far behind the DnD-Transformer in rich-text generation tasks.
>
> > W4: I did not find sufficient evidence in the experiments to convincingly show a reduction in VQ loss.
> >
>
> In our paper, Figure-3 and Table-1 show that with more depth of the RQ-Tokenizer, the VQ loss (L2 loss) becomes smaller and result in better image reconstruction performance both in natural and rich-text images.
>
> > W-others:
> >
>
> Thanks for your mention, we added the results of MAR[1] to the main table and add to the related work. The reconsturction result are shown in Figure-3 and Table-1 to validate that VQ with more depth would lead to better image reconstruction.
>
> We hope our response resolves your concerns and to hear your feedback. If you have further questions, we are always here to respond. Thank you again.

---

> ### Comment · Reviewer_4s2g · 2024-11-23
>
> Thanks for your reply.
>
> W1 and W4’s responses still leave some questions unanswered. While I understand that the authors’ work builds on the follow-up research of RQ, they have not provided a clear rationale for why a single transformer might outperform RQ’s two independent prediction modules. Admittedly, this issue does not diminish the overall value of the paper.
>
> Regarding W4, my original question was whether the proposed method improves reconstruction compared to the baselines, such as RQ or VQ. The authors have clarified in their response to W1 that their approach does not aim to enhance Visual Tokenizers. However, I noticed that Section 2.2 repeatedly references the DnD tokenizer. Could the authors elaborate further on its role, e.g., can I expect a better Table 1 with baseline RQ?
>
> As for W2, I find the response somewhat insufficient. I was hoping for examples demonstrating that images generated using DnD genuinely achieve Vision-Language Intelligence, as suggested by the title. Specifically, I expected DnD to enable the text within the generated rich-text images to be coherent and meaningful. However, Fig. 7 does not align with these expectations. Could the authors provide further evidence to substantiate their claim in Section 4.4 of “A Spark of Vision-Language Intelligence”?
>
> I thank the author for providing additional results for W3.

---

### Official Review · Reviewer_3Gvw · 2024-11-04

**Soundness:** 3
**Presentation:** 3
**Contribution:** 3
**Rating:** 8
**Confidence:** 4

**Summary:**

The paper proposes a 2-Dimensional Autoregressive Transformer (DnD-Transformer) which, essentially, adds iterative refinement to visual token generation in the autoregressive regime. This iterative refinement is reminiscent of diffusion model inference (but not the same).

**Strengths:**

We know that iterative refinement is currently key in methods to generate images. For example, diffusion and rectified flow models which are SoTA have this iterative refinement quality. Embedding iterative refinement in the AR transformer block is a very interesting idea. Further, the paper presents ablations that show monotonically improving FID and other scores when the depth is higher, which is consistent with this claim. Also, they present results that rival SoTA diffusion models for class-to-image generation, which is very promising. The images they show in the paper are also convincing, with very high quality - and the interesting phenomenon of coherence in text generation even when trained without text-labels. Overall it's an interesting paper with a strong directive idea.

**Weaknesses:**

1. The paper needs objective data on FLOPs, training time, and inference speed. This is very important since adding the DnD-Transformer layers increases all of these. It needs comparisons to SoTA diffusion, rectified flow, and AR models. This is very important for judging the paper since increasing computation has to be balanced out by increased performance - and both need to be presented, not just one.
2. Only explores class-to-image and unconditional image generation.

**Questions:**

- My biggest question is on computational complexity. We need more clarification on the additional compute needed for different depths of DnD-Transformer layers, for both inference and training, and comparisons to other work.
- Is there any reason text-to-image is not explored?
- Can the method generalize to higher resolutions without exploding complexity?

My final score will depend on these clarifications.

---

> ### Author Response · Authors · 2024-11-21
> **Response to Reviewer 3Gvw**
>
> Dear Reviewer 3Gvw,
>
> Thanks for your detailed and valuable comments and suggestions. It is encouraging to see that you find our work interesting and with strong directive idea. We are here to address all your concerns:
>
> > W1 & Q1: objective data on FLOPs, training time, and inference speed
> >
>
> Thanks for your mention, we have added the detailed benchmark on training/inference complexity compared to different baselines (Table-5). All experiments are conducted on A100 GPUs.
>
> | Imagenet(256x256) | Total Training FLOPs | Num-Parameters | gFID | Inference-Time(second/image) | Training-Time (mintues/epoch) |
> | --- | --- | --- | --- | --- | --- |
> | DnD-Transformer(depth-2) | 1.02x (2.57e17) | 1.44 B | 2.58 | 4.23s | 23min |
> | DnD-Transformer(depth-3) | 1.04x (2.62e17) | 1.45 B | 2.53 | 4.48s | 25min |
> | LlamaGen(depth-1) [1] | 1x (2.52e17) | 1.43 B | 4.12 | 4.05s | 22min |
> | DiT [2] | - | 675M | 2.27 | 15.45s (100 steps) | - |
> | SD3 [3] | - | 3B | - | 10.78s (28 steps) | - |
>
> Actually, DnD-Transformer only adds additional prediction head (linear projection) to the backbone transformer model. Assume the dimension  of hidden state of the LLM is H, the vocab size is V, and there are N additional output heads. DnD-Transformer only adds HxVxN parameters, which is only a very small portion of parameters comparing to the LLM backbone according the above table. During training and inference, only N additional linear projection and softmax operation are conducted.
>
> > W2 & Q2: Only explores class-to-image and unconditional image generation.
> >
>
> Our main baseline LlamaGen[1], utilizes 50M Laion-coco + 10M **internal** data to conduct text-2-image generation experiments. We are unable to get internal data and train on such a large dataset due to limited computation resources. We think the ImageNet and Rich-Text image generation can already show the effectiveness of our method compared to the LlamaGen baseline.
>
> > Q3: generalize to higher resolutions without exploding complexity?
> >
>
> The answer is yes. As mentioned in W1&Q1,  compared to the baseline AR generation LlamaGen, DnD-Transformer only adds HxVxN parameters to the backbone model and only N additional linear projection and softmax operation are conducted, which are irrlevent of the image input resolution.
>
> [1] Autoregressive Model Beats Diffusion: Llama for Scalable Image Generation
>
> [2] Scalable Diffusion Models with Transformers
>
> [3] Scaling Rectified Flow Transformers for High-Resolution Image Synthesis
>
> We hope our response resolves your concerns and to hear your feedback. If you have further questions, we are always here to respond. Thank you again.

---

### Official Review · Reviewer_FQZc · 2024-11-04

**Soundness:** 2
**Presentation:** 3
**Contribution:** 3
**Rating:** 6
**Confidence:** 4

**Summary:**

In this work, the authors introduce DnD Transformer, which predicts discrete image tokens in both sequence and depth direction. In sequence direction, the model is trained to generate tokens one by one as standard AR. In depth direction, the model generate one complete token in a residual-quantized manner. Experiments on ImageNet show competitive performance against baseline AR models. The model also show promising results on text-heavy image generation tasks.

**Strengths:**

1. The work investigates an interesting problem about improving autoregressive image generation on discrete tokens.
2. The model achieves competitive performance on standard ImageNet benchmark.
3. The model demonstrates capability of generating images with rich text and graphical elements.

**Weaknesses:**

1. Some implementation details are missing which affect the evaluation of proposed method.
2. Some baselines are missed on standard ImageNet generation tasks.
3. The model claims it shows a spark of vision-language intelligence by showing results on the rich-text image generation. However, it's a bit unclear to me whether these results lead to actual language understanding. It may beyond the scope of this work but it would be interesting to see how it performances in vision-language understanding tasks. The authors may consider being careful about the claim.

Please find more details in Questions section below.

**Questions:**

1. It's not very clear about the gain from depth-wise autoregression. It would be good to have results with depth 1 or 3 on ImageNet reported as well. Also on rich-text image generation, DnD is trained with depth 1. How would the performance change if more depths are applied?
2. There're some AR-based image generation tasks that are not included in ImageNet256 benchmark. For example, VAR [1] curates a resolution-changing tokenizer for image generation.
3. For rich-text image generations, how is SD3 implemented? Is it re-trained on tokenizer trained on rich-text datasets or finetuned with its original tokenizer? More details would help understand the performance gap between SD3 and DnD.
4. Also, are there quantitative evaluations for rich-text image generation, like the rOCR used in evaluation of reconstruction performance?
5. The tokenizers are trained separately for ImageNet and rich-text datasets. Will tokenizers trained on merged datasets hurt the performance on either side?

References:

[1] Visual Autoregressive Modeling: Scalable Image Generation via Next-Scale Prediction, https://arxiv.org/abs/2404.02905

---

> ### Author Response · Authors · 2024-11-21
> **Response to Reviewer FQZc (1/2)**
>
> Dear Reviewer FQZc,
>
> Thanks for your detailed and valuable comments and suggestions. It is encouraging to see that you find our work interesting and acknowledge our achievements. We are here to address all your concerns:
>
> > Weakness 1: Some implementation details are missing
> >
>
> If we understand you right, you are mentioning the details in Question 3. Please refer to Q3 for our response and feel free to raise other questions about implementation details.
>
> > Weakness 2: Some baselines are missed
> >
>
> Thanks for your advice. We have added VAR to the main table and related work.
>
> > Weakness 3: About more vision-language ability
> >
>
> We have observed some interesting phenomena in our experimental rich-text generation experiments, as shown in Figure 20 in the appendix. The DnD-Transformer, when trained to generate pure text images, exhibits a generation pattern similar to LLMs. It can understand the names of celebrities such as Barack Obama and knows he is related to the United States, but it also tends to repeat and hallucinate. We agree that it is worth exploring more vision-language tasks in the future.
>
> > Question 1 : About depth vs performance
> >
>
> We conduct the gFID vs. depth experiments on PIL-Text and Imagenet datasets.  The results are shown below.
>
> We conduct experiments on the same Transformer-XXL structure with the distance of each code-output layer to 3.
>
> | PIL-Text(512) | depth=1 | depth=2 | depth=8 | DDPM |
> | --- | --- | --- | --- | --- |
> | rOCR | 0.73 | 0.81 | 0.83 | - |
> | PPLocr  (lower is better) | 312 | 298 | 295 | 1645 |
>
> | ImageNet256x256 | depth=1 | depth=2 | depth=3 | depth=4 |
> | --- | --- | --- | --- | --- |
> | rFID | 2.98 | 0.93 | 0.63 | 0.6 |
> | gFID | 4.12 | 2.58 | 2.53 | 4.79 |
>
> With increased depth, the PIL-Text generation result could be further improved. For imagenet, as we discussed about it in section 4.5 in the original paper that the increase in depth does not always bring in better generation quality in our experiments. The result on ImageNet256 shows that as the depth of code increases, the reconstruction performance keeps improving, however the relative improvement become marginal as the depth goes deeper. The generation performance first improves (from 1 to 3) then degrades (from 3 to 4). We think the result is due to the marginal improvement in rFID when the depth is large. Within the same backbone model, the overall prediction complexity and error accumulation becomes larger , but the theoritical performance improvement is marginal (rFID) when the depth grows.

---

> > ### Author Response · Authors · 2024-11-21
> > **Response to Reviewer FQZc (2/2)**
> >
> > > Question 2 : Important AR-based image generation baselines
> > >
> >
> > Thanks for your mention, we have added  VAR to the main table and related work.
> >
> > > Question 3:  Implementation details of SD3
> > >
> >
> > We do not retrain the tokenizer of SD3 as we found that it actually has better text-image reconstruction ability on both PIL-Text and Arxiv images than our RVQ tokenizer, as shown in Table 3. This means it can better reconstruct the text information in the image. We believe that they are not only trained on non-text images based on these results (the SD3 paper does not report the training dataset). However, even with such a strong tokenizer, diffusion models still lag behind DnD-Transformer when trained to generate rich-text images. This evidence better justifies our claim that DnD-Transformer outperforms LDM on this task.
> >
> > > Question 4: quantitative evaluations for rich-text image generation
> > >
> >
> > Yes, we have proposed PPL_ocr to quantitatively evaluate the text quality in the generation results, as explained in Section 4.1 of the original paper, and the results are shown in Figure 6b. We first run OCR on the generated rich-text images (PIL-Text) and use an off-the-shelf LLM to compute the PPL of the OCRed text. A higher PPL_ocr means the LLM allocates less likelihood to the generated text, indicating a worse generated image. We find that DnD-Transformer achieves significantly lower PPL_ocr than its diffusion counterpart.
> >
> > > Question 5: Will tokenizers trained on merged datasets hurt the performance on either side?
> > >
> >
> > Thanks for your mention,  we find that the RQ tokenizers(8-depth) trained on the merged dataset would **not** hurt the performance on either side, the experiments result are shown below. We think the reason is that relatively large codebook size has enough space for encode both the natural image and text image information.
> >
> > | Training Dataset | Imagenet | PIL-512 | ImageNet+PIL-512 |
> > | --- | --- | --- | --- |
> > | rFID on ImageNet | 0.42 | - | 0.41 |
> > | rOCR on PIL text | - | 0.83 | 0.83 |
> >
> > We hope our response resolves your concerns and to hear your feedback. If you have further questions, we are always here to respond. Thank you again.

---

### Official Review · Reviewer_BF3N · 2024-11-05

**Soundness:** 2
**Presentation:** 2
**Contribution:** 2
**Rating:** 6
**Confidence:** 4

**Summary:**

This work addresses the information loss bottleneck in vector-quantized (VQ) autoregressive image generation by introducing a novel model architecture, the 2-Dimensional Autoregression (DnD) Transformer. The DnD Transformer is an end-to-end model capable of generating higher-quality images while maintaining the same backbone model size and sequence length, presenting a fresh optimization perspective for autoregressive image generation. Experiments on ImageNet and rich-text images demonstrate that the DnD Transformer successfully generates detailed, high-quality images, highlighting its potential for advancing vision-and-language models with a new approach.

**Strengths:**

It’s intriguing to see the potential for training a vision-and-language model solely on images. When training such models using textbooks or other materials, preprocessing to associate images with the relevant text is often challenging. Training an LMM purely on images might offer a more principled and human-like learning approach.

**Weaknesses:**

#1. Compared to the RQ-Transformer paired with RQ-VAE, what are the benefits of this work? The proposed method appears largely similar, with no observed improvements over RQ-VAE.

#2. I believe the manuscript requires extensive revision. Some references are incomplete, with some citing Wikipedia, which is not a valid source. Furthermore, the comparison between DnD-Transformer and existing approaches, especially RQ-Transformer, lacks clarity and is not fully convincing.

#3. I believe many experiments in this paper may lead to misinterpretation, as the comparisons are not conducted under fair conditions. I suggest that the authors address my questions in #5 and #8 to ensure a more accurate comparison.

**Questions:**

#1. I'm not trying to pinpoint this assertion, but I don't personally buy this argument, since autoregressive image generation research was already very popular before the release of ChatGPT. Additionally, by 2022, the research community had already shifted to diffusion models. GLIDE (2021), Latent Diffusion (2022), DALL-E 2 (2022), and Imagen (2022) were all published before ChatGPT.

#2. The claim in the phrase “a spark of vision-language intelligence for the first time” appears to be overstated, especially given the limited scope of the experiments conducted. It is notable that AR-based image generation demonstrates strong performance in text rendering, particularly for documents, compared to diffusion models. However, can this really be considered a spark in vision-and-language intelligence?"

#3. How is the ICR of JPEG computed? Additionally, it would be preferable to include a proper reference from a primary source or technical documentation rather than Wikipedia.

#4. In section 2.2, the authors discuss the differences between the proposed approach and RQ-VAE, but I find the explanation somewhat unconvincing. Could you elaborate on the distinctions and explain why the newly proposed component might be expected to outperform RQ-VAE?

#5. In Table 1(a), code usage is reported at 100%. Did you apply any specific techniques to enhance code usage, such as restarting dead codes or similar methods?

#6. In Table 1(b), SDXL and SD3 are trained on ImageNet and should be considered zero-shot tokenizers. However, this isn’t indicated in the table, which leads to misinterpretation of the results.

#7. As I understand it, unlike RQ-Transformer, DnD-Transformer predicts codes along the depth dimension simultaneously. However, HQ-Transformer, a follow-up to RQ-Transformer, also explores predicting codes in this way. If my understanding is correct, it might be beneficial to include HQ-Transformer as a baseline for comparison.

#8. Regarding Figure 3, which dataset was used for the tokenizer in SD3? Did you use the original SD3 tokenizer? If so, I question whether this is a fair comparison, as open-source image generation models generally do not include OCR images in their datasets. If the claim is that DnD-Transformer performs well in generating rich-text images, it would be more appropriate to train a tokenizer specifically on rich-text images. I suggest training an LDM, including a tokenizer (such as continuous VAE f8), on rich-text images and then comparing DnD-Transformer with LDM under these conditions. In that case, I am not certain that DnD-Transformer would outperform LDM.

---

> ### Author Response · Authors · 2024-11-21
> **Response to Reviewer BF3N (1/2)**
>
> Dear Reviewer BF3N,
>
> Thanks for your detailed and valuable comments and suggestions. It is encouraging to see that you find our work intriguing and potentially offering a more principled and human-like learning approach. We are here to address all your questions, clarify some misunderstandings and fix the overstated statements.
>
> > Weakness 1:  "Advantage over with RQ-VAE"
> >
>
> DnD-Transformer does not improve RQ-VAE as a visual tokenizer. Instead, it uses the RQ-VAE as the tokenizer. Its biggest advantage over the RQ-VAE paper is that it is a single end-to-end transformer that predicts the multicodes without expanding the sequence length, unlike the RQ-Transformer, which uses two individual transformers to predict the codes. This simple yet effective structure is more efficient in both training and inference, and it can be better integrated with current LLM structures without involving additional models. The novelty is also acknowledged by reviewers Zh5x, FQZc, 3Gvw, and 4s2g.
>
> > Weakness 2: "I believe the manuscript requires extensive revision"
> >
>
> Thanks for your advice. We have changed the citation to official publications and explain the difference between DnD-Transformer and RQ, HQ-Transformer. We will further explain these in our response to question #3 and #4.
>
> > Weakness 3:  "I believe many experiments in this paper may lead to misinterpretation"
> >
>
> The comparison is fair since the VAE of SD3 actually has better text reconstruction ability than our trained RQ-VAE, according to Table 3. The rOCR of the SD3 VAE is better than our best 8-layer RQ-VAE trained on rich-text images. However, even with a better visual tokenizer, diffusion models still lag behind our DnD-Transformer in the rich-text generation task. This evidence better justifies our claim that DnD-Transformer outperforms LDM on this task. We use the same VQ training pipeline as RQ-VAE without changing the database. We will further explain these points in our response to questions #5 and #8.
>
> > Question 1: Wording problem
> >
>
> We think there might be some misunderstanding about our wording. We use the word "resurgence" to imply that AR image generation methods like DALL-E1, Muse, and RQ-Transformer were popular, as you mentioned, "before ChatGPT," but were later replaced by diffusion models such as DALL-E2 and Stable Diffusion. However, recently (after ChatGPT), there has been a tremendous amount of work, such as Chameleon[1] , Emu3[2], which use more advanced LLM structures such as LLaMA to conduct AR image generation tasks. This is why we call it a "resurgence".
>
> > Question2: "The claim in the phrase “a spark of vision-language intelligence for the first time” appears to be overstated"
> >
>
> Thanks for your mention. We have removed the claim from the paper and replaced with more accurate "it shows that we can conduct accurate language modeling with pure image generation model."
>
> > Question3: "How is the ICR of JPEG computed?"
> >
>
> ICR of JPEG was calculated by the ratio between number of bits used to store the compressed image and the original image. As JPEG has an additional parameter to control the compress rate, (1-100, higher means less information loss), we use the common compression rate reported by [3] and report the ICR of 5%~10%, which is significantly higher than that of VQ. We have fixed it in the paper.

---

> > ### Author Response · Authors · 2024-11-21
> > **Response to Reviewer BF3N (2/2)**
> >
> > > Question4: "explanation on the difference between RQVAE"
> > >
> >
> > We adopt RQ-VAE as the multi-layer code tokenizer. DnD-Transformer does not propose a new image tokenizer. It proposes a new autoregressive manner to predict the multi-layer code in an end-to-end model. It is not supposed to outperform RQ-VAE in image reconstruction, and we also did not claim this in the paper.
> >
> > > Question5:  "code usage is reported at 100%."
> > >
> >
> > We completely follow RQVAE as the multi-layer code tokenizer and use the same training pipeline. They use random restart of unused codes during training. We would add this in the paper.
> >
> > > Question6: "SDXL and SD3 are trained on ImageNet and should be considered zero-shot tokenizers"
> > >
> >
> > Thanks for mention, we would mark this on the paper. However, we do not see any public details that tells what image datasets the SDXL and SD3's tokenizers are trained on, according to the public paper of SDXL[4] and SD3[5]. The claim that "SDXL and SD3 are trained on ImageNet" might be not true. However, it is very likely(almost for sure) that they also train the VAE on rich-text images. Otherwise, they should not be able to reconstruct the detailed texts better than our trained RQ tokenizers according to table-3b.
> >
> > > Question7: "As I understand it, unlike RQ-Transformer, DnD-Transformer predicts codes along the depth dimension simultaneously. "
> > >
> >
> > Thank you for your mention, we have added the results of HQ-Transformer.  Both RQ-Transformer and HQ-Transformer use **multiple** separate transformer models to predict the multi-layer codes. DnD-Transformer just uses **one** transformer decoder to predict the multi-layer code. DnD-Transfomer **does not predict the code simultaneously,** it actually predicts the code in depth direction also in an autoregressive manner according to figure-5 (each layer outputs one code in the GPT model). DnD-Transformer predicts code of different depth in one forward pass, which makes the training and inference more efficient.
> >
> > Additionally, we have added the detailed benchmark on the training/inference budgets.
> >
> > | Imagenet(256x256) | Total Training FLOPs | Num-Parameters | gFID | Inference-Time(second/image) | Training-Time (mintues/epoch) |
> > | --- | --- | --- | --- | --- | --- |
> > | DnD-Transformer(depth-2) | 1.02x (2.57e17) | 1.44 B | 2.58 | 4.23s | 23min |
> > | DnD-Transformer(depth-3) | 1.04x (2.62e17) | 1.45 B | 2.53 | 4.48s | 25min |
> > | LlamaGen(depth-1) | 1x (2.52e17) | 1.43 B | 4.12 | 4.05s | 22min |
> > | DiT | - | 675M | 2.27 | 15.45s (100 steps) | - |
> > | SD3  | - | 3B | - | 10.78s (28 steps) | - |
> >
> > DnD-Transformer only  adds additional prediction head (linear projection) to the backbone transformer model. Assume the dimension  of hidden state of the LLM is H, the vocab size is V, and there are N additional output heads. DnD-Transformer only adds HxVxN parameters, which is only a very small portion of parameters comparing to the LLM backbone according the above table. During training and inference, only N additional linear projection and softmax operation are conducted.
> >
> > > Question8: "Regarding Figure 3, which dataset was used for the tokenizer in SD3?"
> > >
> >
> > The tokenizer of SD3 actually has better text-image reconstruction ability on both PIL-Text and Arxiv images than our RVQ tokenizer, as shown in Table 3b. This means it can better reconstruct the text information in the image and we think it is not necessary to retrain the tokenizers since they are already stronger than our trained models. We believe that they are not only trained on non-text images based on these results (the SD3 paper does not report the training dataset). However, even with such a strong tokenizer, latent diffusion models still lag behind DnD-Transformer when trained to generate rich-text images. This evidence better justifies our claim that DnD-Transformer outperforms LDM on this task.
> >
> > [1] Chameleon: Mixed-Modal Early-Fusion Foundation Models
> >
> > [2] Emu3: Next-Token Prediction is All You Need
> >
> > [3] The JPEG still picture compression standard
> >
> > [4] SDXL: Improving Latent Diffusion Models for High-Resolution Image Synthesis
> >
> > [5] Scaling Rectified Flow Transformers for High-Resolution Image Synthesis
> >
> > We hope our response resolves your concerns and to hear your feedback. If you have further questions, we are always here to respond. Thank you again.

---

> > ### Comment · Reviewer_BF3N · 2024-11-26
> >
> > I apologize for joining the discussion so late. Thank you for your detailed response. My original questions regarding this part have been resolved. I had misunderstood the difference between the DnD-Transformer and the RQ-Transformer. I have no further questions.

---

> > > ### Comment · Reviewer_BF3N · 2024-11-26
> > >
> > > In addition, I made a mistake regarding Question 6. I initially thought that the authors had re-trained the SDXL and SD3 tokenizer on ImageNet.
> > >
> > > After considering the reviews from my fellow reviewers and reflecting on the rebuttal process, I have become more positive about this work. As a result, I have increased my score to 6.

---

### Official Review · Reviewer_Zh5x · 2024-11-07

**Soundness:** 4
**Presentation:** 4
**Contribution:** 3
**Rating:** 6
**Confidence:** 3

**Summary:**

This paper addresses the problem of image modeling and generation using discrete latent representations and autoregressive (AR) sampling. The authors note that larger codebooks are need to improve visual quality but scaling up the codebook directly is difficult. They suggest a using multiple codes per token and adopt residual vector quantization (RVQ or just RQ) based on an earlier model called the RQ-Transformer.

They then present a creative approach for predicting the residual codes by adding a prediction head between different layers within the normal multi-layer transformer. This approach adds only a trivial amount of extra parameters and computational cost, and requires only minor changes to existing LLMs. Contrast with the RQ-Transformer that adds a new (albeit small) transformer, which is harder to integrate into existing LLM code. Through empirical validation, the authors show that their "DnD-Transformer" yields good generation results compared to RQ-Transformer, LlamaGen, and other methods.

Using RVQ to decompose a large codebook is not new, but adding prediction heads between the layers of the LLM transformer is. The authors show that this approach outperforms parallel prediction and vertical prediction (visualized in Fig. 5), two other approaches for predicting multiple codes from a single forward pass.

In addition to generation on ImageNet-256 evaluated using standard metrics (gFID, IS, etc.), the authors also introduce rOCR that evaluates an autoencoder in terms of how well text in a reconstructed image can be recognized. They show that larger depth values (more residual codes) improves this metric, as expected (see Table 1b). For generation, they measure perplexity using Qwen2.5-72B over text extracted with Qwen2-VL-72B and show that their approach performs better that a diffusion-based baseline (Fig. 6b). They argue that AR prediction is better suited to generating images of text compared to the "simultaneous" generation of diffusion models. This is where the "spark of vision-language intelligence" comes from: the DnD-transformer is trained on images of text and can then generate images of text with a few recognizable words and short phrases (see Fig. 7).

**Strengths:**

The main strength of the paper is the authors' creative solution to predicting multiple codes per token in an efficient and effective manner.  The approaches used in other works all have significant drawbacks:
 - use more tokens, which is computationally expensive
 - use a second transformer as in RQ-Transformer, which complicates the model, especially if you're building on top of an existing LLM
 - grow the codebook, which is memory-limited for VQ (though not FSQ or binary quantization as in MagVit v2 and MaskBit) and makes prediction more difficult
 - use multiple codes and predict them in parallel, which doesn't work very well (also discussed in this paper).

**Weaknesses:**

I see two main weaknesses in the paper. First, I'd like to see more evaluation of the DnD-Transformer in terms of sensitivity to layer indexes for the prediction heads both within an otherwise fixed architecture (e.g., stick with DnD-Transformer-XXXL and vary the indexes) and for smaller/larger models (e.g., if #heads == #layers, the smallest option, does the approach still work?). While the layer indexes are "just" a hyperparameter, it would be quite interesting to know if the approach is relatively robust to the layer choices or if there's a pattern. For example, for a fixed number of layers (and thus compute), you can ask how that compute should be used: do you want more layers before the first code is predicted? An even distribution across codes? Maybe the first code is relatively easy and you need successively more compute for later codes.

The second weakness deals with evaluation of generation results as a function of depth. If I'm interpreting the paper correctly, the ImageNet results in Table 2 use depth=2 (see the caption for Table 2), and the text-image generation results use depth=1 (see Section 4.4). Note that depth=1 is the baseline where no residuals are used (i.e., it's just an standard next-token predictor with one code per token). These low depth values seem to undermine the core contribution of the DnD-Transformer.

Note that the "Generation Results on arXiv-Image" sub-section does say that an "8-layer visual tokenizer and corresponding DnD-transformer..." If that means that depth=8, then I'm less concerned. Nonetheless, I'd like to see a chart showing gFID vs. depth (much like other VQVAE-based papers show gFID vs. codebook size). Figure 1b has the right structure, but it shows *reconstruction* metrics, which should always improve as depth increases. This does not always translate to generation quality.

**Questions:**

I'm not sure how to interpret the code usage reported in Table 1a vs. Figure 4a. The table shows 100% usage up to depth=8, but the chart shows code usage falling off with depth. Is this a mistake or are they showing something different?

A minor point (that's not actually a question): In Section 2.1 the paper says "each code has log N bits [of] information". It should be "log_2 N bits" or "log N nats".

---

> ### Author Response · Authors · 2024-11-21
> **Response to Reviewer Zh5x (1/2)**
>
> Dear Reviewer Zh5x,
>
> Thanks for your detailed and valuable comments and suggestions. It is encouraging to see that you find our work creative and have significant advantage over other methods. We are here to address your concerns with our updated experiments results and clarification:
>
> > Weakness1: Evaluation of the DnD-Transformer in terms of sensitivity to layer indexes
> >
>
> We have incorporated the results of DnD-Transformer-XXL using various output layer indices, while maintaining a 2-layer configuration. The final code output layer is consistently aligned with the last transformer layer, while we vary the layer index for the initial code output layer. The table below illustrates the relationship between the distance of these two code output layers and the resulting FID score for generation on ImageNet256
>
> | The first layer's distance to the second layer | 1 | 3(original) | 5 | 7 | 9 | 12 | 15 | 24 |
> | --- | --- | --- | --- | --- | --- | --- | --- | --- |
> | gFID | 5.12 | 2.58 | 2.53 | 2.66 | 2.54 | 2.64 | 3.59 | 4.59 |
>
> The conclusion drawn from our study indicates that the DnD-Transformer is relatively robust across a broad spectrum of layer indices, ranging from 3 to 12, with only minor variations in gFID scores (between 2.53 and 2.64). However, when the gap between the layers is too narrow or too wide, the results tend to diminish. The XXL model comprises 48 layers in total. Our findings suggest that more layers should be allocated for predicting the first layer of code, indicating that the first-layer code is more challenging to predict than the later-layer code. This conclusion aligns with our analysis of code distribution, as shown in Figure 4(a, b). The image demonstrates that the first layer code utilizes a broader range of codes (Figure a shows the first layer employs 100% of the entire codebook, whereas the second layer uses approximately 70%) and displays a higher variance in code selection (Figure b). This broader usage and higher variance suggest that the first-layer code's distribution is more complex than that of the second layer, thereby increasing the prediction difficulty. Consequently, more layers are needed to predict the first code than the second code.
>
> Additionally, when we explored the configuration where the number of heads equals the number of layers, we observed that the generation quality considerably deteriorated under this setup (with gFID scores of 23.55 for #layers=1 and 26.79 for #layers=2). We believe the cause is linked to the necessity of having more layers to accurately predict the first code, which in turn enhances the overall generation quality. In comparison to the aforementioned results, when 46 additional layers are allocated for predicting the first-layer code, the gFID improves significantly from 26.79 to 5.12.

---

> > ### Author Response · Authors · 2024-11-21
> > **Response to Reviewer Zh5x (2/2)**
> >
> > > Weakness2: Evaluation of generation results as a function of depth
> > >
> > 1. Why different depth for imagenet and text-image? Clarification on "Generation Results on arXiv-Image"
> >
> > We want to clarify this point. We conduct two text-image generation experiments, the PIL-Text and arXiv image. We train RQ-tokenizers with different depths (1,2,8) and find that the arXiv image requires a depth-8 tokenizer to reconstruct the language in the image due to the various fontstyle and more complex figure-text relationship (according to Figure3), the PIL-Text can be well reconstructed with a depth-1 tokenizer. So we run the generation experiments with a depth-8 DnD-Transformer for arXiv image (Yes, the arxiv-image generation experiment was conducted with depth-8 tokenizer and depth-8-DnD-Transformer), depth-1 for PIL. The depth-1 PIL experiments can well reveal AR's significant advantage over diffusion in plain text generation(PIL), we conduct additional experiments showing that with increased depth, the PIL-Text generation result could be further improved according to the table below. However, more code depths and DnD-Transformer are needed to generate complex rich-text images (1-depth AR could not achieve this).
> >
> > | PIL-Text(512) | 1 | 2 | 8 | DDPM |
> > | --- | --- | --- | --- | --- |
> > | rOCR | 0.73 | 0.81 | 0.83 | - |
> > | PPLocr  (lower is better) | 312 | 298 | 295 | 1645 |
> > - gFID vs. depth on Different Benchmarks
> >
> > We conduct the gFID vs. depth experiments on PIL-Text and Imagenet datasets.  The results of PIL-text experiment are shown above, the imagenet's is shown below.  We discussed about it in section 4.5 in the original paper that the increase in depth does not always bring in better generation quality in our experiments. We conduct experiments on the same Transformer-XXL structure with the gap of code-output layer to 3. We list the concrete results below:
> >
> > | ImageNet256x256(cfg=2) | 1 | 2 | 3 | 4 |
> > | --- | --- | --- | --- | --- |
> > | rFID | 2.98 | 0.93 | 0.63 | 0.6 |
> > | gFID | 4.12 | 2.58 | 2.53 | 4.79 |
> >
> > The result on ImageNet256 shows that as the depth of code increases, the reconstruction performance keeps improving, however the relative improvement become marginal as the depth goes deeper. The generation performance first improves (from 1 to 3) then degrades (from 3 to 4). We think the result is due to the marginal improvement in rFID when the depth is large. Within the same backbone model, the overall prediction complexity and error accumulation becomes larger , but the theoritical performance improvement is marginal (rFID) when the depth grows.
> >
> > > Question1: Interpretation of the code usage reported in Table 1
> > >
> >
> > This is not a mistake. Different code prediction layers share the same codebook. Table 1a shows the overall code usage calculated over all depths, and Figure 4a shows the code usage of different layers. This means that different layers actually use different parts of the code, but they use 100% of the codebook. For example, if the total codebook is (a, b, c, d), layer 1 uses (a, b, c), and layer 2 uses (c, d), then the overall usage is 100%, layer 1 usage is 75%, and layer 2 usage is 50%.
> >
> > > Question2: A Minor point on mathematical notation
> > >
> >
> > Thank you, we have fixed it in the paper.
> >
> > We hope our response resolves your concerns and to hear your feedback. If you have further questions, we are always here to respond. Thank you again.

---

### Meta-Review · Area_Chair_4Jet · 2024-12-19

**Metareview:**

The paper introduces a new formulation for vector-quantized autoregressive image generation that overcomes complications of existing 1D approaches (namely information loss and computational complexity) by predicting tokens that are expanded depth-wise. This gives rise to a 2D sequence for modeling image content where the first dimension is spatial and the second dimension encodes visual content at progressive levels of depth/detail. By predicting tokens at multiple depths simultaneously, the method reduces the complexity relative to the standard approach of increasing the sequence length, while increasing the fidelity/compression ratio. The approach is validated by a series of experiments comparing to existing state of the art approaches, both for conditional and unconditional image generation. In terms of conditional image generation, the method improves significantly in FID over existing AR baselines. More interestingly, the method does well at unconditional image generation where the images contain fine grained text, hinting that the model is capable of generating text with a sophistication previously reserved for standard language models.

Strengths:
The approach is well motivated by referring to the compression view of vector quantization and the resulting complications. The 2D solution is elegant and simple, and seems to work well in practice, including rivaling state of the art diffusion approaches. The idea of training a joint image-text model purely with images is a potentially exciting direction motivating early fusion approaches. Analyses and ablations confirm the method and demonstrate efficiency gains.


Weaknesses:
Some concerns that the sparks of vision language intelligence may be a bit overclaimed, though authors softened these claims. Also, further analysis/theory could illuminate why the single-stage parallel prediction of depth might improve over two independent precictions as in the original RQ work.

Decision reasoning:
This paper is well motivated and backed up by strong empirical results. Reviewers were generally positive about the paper and one noted that it is worthy of highlighting the contribution. I agree that this work deserves visibility particularly due to its practical value in improving fidelity without significant increased computational overhead, and I think the community can build on the work to better undestand why a single stage predictor is so effective and to consider additional variants of the idea.

**Additional Comments On Reviewer Discussion:**

Most questions asked by reviewers were addressed adequately by the authors, and the one reviewer with borderline rejection score never responded to the authors’ final rebuttal, which seemed adequate to me.

---

### Decision · Program_Chairs · 2025-01-22

Accept (Poster)